# Spectral Bias in Practice: the Role of Function Frequency in Generalization

**Sara Fridovich-Keil** [*]
University of California, Berkeley
sfk@eecs.berkeley.edu

**Raphael Gontijo-Lopes**
Google Brain
iraphael@google.com

**Rebecca Roelofs**
Google Brain
rofls@google.com

## Abstract

Despite their ability to represent highly expressive functions, deep learning models seem to find simple solutions that generalize surprisingly well. Spectral bias – the tendency of neural networks to prioritize learning low frequency functions – is one possible explanation for this phenomenon, but so far spectral bias has primarily been observed in theoretical models and simplified experiments. In this work, we propose methodologies for measuring spectral bias in modern image classification networks on CIFAR-10 and ImageNet. We find that these networks indeed exhibit spectral bias, and that interventions that improve test accuracy on CIFAR-10 tend to produce learned functions that have higher frequencies overall but lower frequencies in the vicinity of examples from each class. This trend holds across variation in training time, model architecture, number of training examples, data augmentation, and self-distillation. We also explore the connections between function frequency and image frequency and find that spectral bias is sensitive to the low frequencies prevalent in natural images. On ImageNet, we find that learned function frequency also varies with internal class diversity, with higher frequencies on more diverse classes. Our work enables measuring and ultimately influencing the spectral behavior of neural networks used for image classification, and is a step towards understanding why deep models generalize well.

## 1 Introduction

Two fundamental questions in machine learning are why overparameterized models generalize and how to make them more robust to distribution shift and adversarial examples. Resolving both of these questions requires understanding how complex our models should be.

For instance, it is thought that overparameterized models generalize well because there are implicit regularizers that constrain the complexity of the learned functions. However, the precise nature of these implicit regularizers, and their importance in practice, remains unclear. As for how to achieve robustness, no consensus has emerged regarding function complexity. On the one hand, Cranko et al. [4] and Leino et al. [24] argue that Lipschitz smoothness of the learned function offers a guarantee of robustness; on the other, Shah et al. [37] and Madry et al. [25] argue that simplicity can be harmful and a robust model must actually be more complex than its non-robust counterpart.

One window into function complexity is spectral bias – the tendency of neural networks to learn low frequency (simple and smooth) functions first, then gradually increase the frequency (complexity) of the learned function as training proceeds. Foundational work in this area has shown theoretical evidence of spectral bias by analyzing convergence rates of neural networks towards functions of different frequencies [1, 35].

---

[*]Work done as an intern and student researcher at Google Brain.

36th Conference on Neural Information Processing Systems (NeurIPS 2022).

In practice, spectral bias is difficult to measure: the most direct method involves taking a Fourier transform with respect to the input, which is impractical to compute due to the high dimensionality of images. Early experimental work focused on low-dimensional synthetic data [1, 2] or used proxy measurements of spectral bias [44], for instance by inserting label noise of various frequencies during training [35]. However, the label noise method is limited to binary classification and involves modifying the training data, making it difficult to disentangle how other changes to the training data (such as augmentation) affect the spectral content of the learned function. It remains an open question whether modern neural networks exhibit spectral bias and what role it plays in generalization.

In this work, we investigate model complexity through the lens of spectral bias, by introducing experimental methods to study the frequency decomposition of the functions learned by modern image classification networks. We find that high-accuracy models learn a function that is high-frequency in regions between different image classes but low-frequency within each class, validating the intuition that a "good" model should have sharp decision boundaries to delineate different classes, but smooth behavior within each class.

**Contributions.** We extend the label noise procedure of Rahaman et al. [35] to enable measuring spectral bias in multi-class classification via label smoothing, and apply this technique to understand the function frequencies present in high-accuracy models on CIFAR-10 [23]. We also introduce a second method for measuring the smoothness of a learned function via linear interpolation between test examples, and apply this method to both CIFAR-10 and ImageNet [8] models. We use this method to probe the effects of the training data on the learned function frequencies, and to distinguish the spectral content in paths between images from the same class ("within-class paths") and paths between images from different classes ("between-class paths").

We find that higher-test-accuracy models demonstrate greater separation in frequency content between these path types, with lower frequencies within-class and higher frequencies between-class. On CIFAR-10, this trend holds as we improve the model architecture, train longer, increase the size of the training dataset, improve the data augmentation strategy, and apply self-distillation. On ImageNet, this trend appears again as we consider models with more diverse architectures.

We also explore the relationship between image frequency and function frequency. By further extending the label noise methodology of Rahaman et al. [35] to study spectral bias in directions of interest through the input space, we find that CIFAR-10 models most readily learn functions of the low image frequencies common in natural images.

Via linear interpolation within ImageNet classes, we find that many models are higher-frequency in more internally diverse classes and lower-frequency in more internally consistent classes.

## 2   Related Work

**Implicit bias.** A common belief is that some form of implicit bias imposed by the training procedure or optimization algorithm may account for the generalization ability of overparameterized neural networks, and accordingly, much research has been directed towards understanding these implicit biases [38, 48, 22, 43, 14, 13, 15, 17, 3, 31, 29, 30, 26, 14, 20].

**Spectral bias.** Spectral bias, also called the frequency principle, is a form of implicit bias with much recent attention including theoretical and experimental approaches [34, 44, 35, 1, 50]. Basri et al. [1] studied spectral bias using a linear model of SGD training dynamics to show that models learn low frequency (simple) functions faster, assuming training data that is distributed uniformly on the hypersphere. Basri et al. [2] extended the analysis to consider nonuniformly spaced training data, and found that learning is faster where samples are denser; if sampling is nonuniform, then during training, the learned function will be higher frequency in regions with denser samples. Rahaman et al. [35] used a more direct analysis to show the same spectral bias toward low frequency functions and also posited that high frequency components of the learned function are most sensitive to perturbations in the model parameters, connecting back to the idea of flat optimization minima [30] or smooth loss landscape [27]. They proposed experimental methods to study spectral bias in image classification, but focused on a binary subset of the relatively simple MNIST dataset [9], with mean square error loss.

Our label smoothing experiments are a direct extension of Rahaman et al. [35] to multiclass classification with the more common cross-entropy loss. Our linear interpolation experiments are somewhat

similar to the recently-proposed experimental methods in Zhang et al. [50], except that we sample along paths between images rather than in regions surrounding each image; this allows for a more global measurement of spectral bias that we apply to a broad and distinct set of training procedures beyond the double descent regime studient in Zhang et al. [50].

**Model sensitivity to image frequency.** While we focus on *function frequency*, prior research aimed to understand model sensitivity to image frequencies. Jo and Bengio [19] found that CNNs are sensitive to Fourier statistics of the training data, even those irrelevant to human viewers. Ortiz-Jiménez et al. [33] studied the image frequency bias induced by using a convolutional architecture, and Ortiz-Jiménez et al. [32] argued that models are sensitive primarily to discriminative Fourier directions in the training data. Yin et al. [46] introduced a procedure to measure the sensitivity of trained models to Fourier image perturbations, and applied it to study the effects of adversarial training and data augmentation. Our work adds to this line of research a study of the interplay between two notions of frequency: the function frequency involved in spectral bias, and the image frequencies present in the data.

# 3 Methodology

The goal of our work is to measure the complexity of neural network functions through the lens of frequency. In low dimensions, measuring the frequency decomposition of a function is straightforward and tractable: evaluate the function at dense, uniform sampling positions and compute its discrete Fourier transform. However, the function of interest in image classification is unavoidably high-dimensional, mapping images (with thousands of pixel values) to object classes. It would be intractable even to collect sufficient samples of this function to compute a discrete Fourier transform, let alone compute the transform itself. Instead, we employ two complementary approaches to measure informative proxies of this frequency decomposition.

## 3.1 Label Smoothing

The core idea for measuring function frequency introduced by Rahaman et al. [35] is to construct a sinusoid over the space of images, and to use that sinusoid as a form of label noise during training. Let $\mathcal{D} = \{(\mathbf{X}_i, \boldsymbol{y}_i)\}_{i=1}^{n_{train}}$ be the training examples and $\mathcal{D}_{\text{valid}} = \{(\mathbf{X}_j, \boldsymbol{y}_j)\}_{j=1}^{n_{val}}$ be the validation examples, with $\mathbf{X}_i$ an image and $\boldsymbol{y}_i$ a one-hot class encoding, where $n_{train}$ is the number of training examples, $n_{val}$ is the number of validation examples, $d$ is the side length of an image (assumed to be square for simplicity), $c$ is the number of color channels, and $M$ is the number of classes, so $\mathbf{X}_i \in \mathbb{R}^{d \times d \times c}$ and $\boldsymbol{y}_i \in \mathbb{R}^M$.

To extend this procedure to the multi-class setting, we add noise of various frequencies to an $M$-dimensional label vector via label smoothing, originally introduced as a regularization approach in Szegedy et al. [39]. Intuitively, label smoothing removes some probability from the correct class and redistributes it equally among the remaining classes. Let $S : \mathbb{R}^{d \times d \times c} \to [0, 1]$ be a noise function that maps an input image $\mathbf{X}_i$ to a scalar value between 0 and 1. We apply label smoothing to each label $\boldsymbol{y}_i$, mapping it to $\bar{\boldsymbol{y}}_i = \boldsymbol{y}_i(1 - S(\mathbf{X}_i)) + \frac{1}{M} S(\mathbf{X}_i)$ to retain a valid probability distribution, as in Szegedy et al. [39] but with different functions $S$. Rather than using label smoothing as a regularizer to improve performance, we use it as a form of variable frequency label noise to study spectral bias. We train from scratch using the original examples $\mathbf{X}_i$ and their smoothed labels $\bar{\boldsymbol{y}}_i$. We evaluate on the validation images $\mathbf{X}_j$, comparing to both their original one-hot labels $\boldsymbol{y}_j$ and smoothed labels $\bar{\boldsymbol{y}}_j = \boldsymbol{y}_j(1 - S(\mathbf{X}_j)) + \frac{1}{M} S(\mathbf{X}_j)$. Our label smoothing experiments use the CIFAR-10 dataset [23], where $n_{train} = 50000$, $n_{val} = 10000$, $d = 32$, $c = 3$, and $M = 10$.

This method is introduced in Figure 1. As training proceeds, we see both the training loss (between predictions and smoothed labels $\bar{\boldsymbol{y}}_i$) and noisy validation loss (between predictions and smoothed labels $\bar{\boldsymbol{y}}_j$) decrease. The clean validation loss (between predictions and one-hot labels $\boldsymbol{y}_j$), however, initially decreases but at some point in training plateaus or begins to increase. At this point, the model has begun to learn the noise function $S$. Accordingly, we introduce **noise fitting**, defined as the difference between the validation loss on one-hot labels and the validation loss on labels smoothed with the same function $S$ that was applied to the training labels: **noise fitting = clean validation loss − noisy validation loss**. We compare different models by computing the minimum noise fitting achieved throughout training. A model with higher **min noise fitting** more readily fits the noise function $S$ than a model with lower min noise fitting.

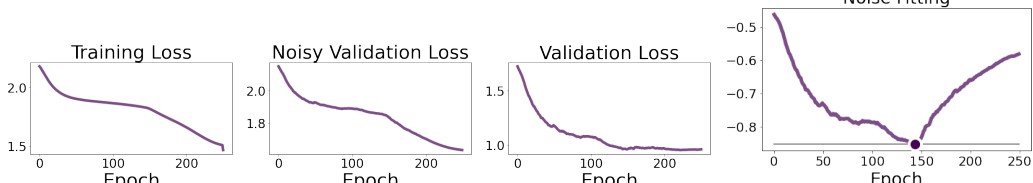

Figure 1: **Noise fitting shows when and how much a network fits a noise function of given frequency.** At roughly epoch 150, noise fitting *(Right)*, the difference between clean and noisy validation loss, exhibits a clear "dip" when the model begins to fit the noise function: training *(Left)* and validation *(Center Left)* loss on the perturbed function drop, while improvement stalls on clean validation data *(Center Right)*. By comparing the minimum value of noise fitting achieved by different models throughout training (the gray line in the right plot), we can compare the relative degree to which different models fit noise functions of varying frequency. A model with higher **min noise fitting** more readily fits the noise function. Here, we train a WideResNet32 model (wide-resnet with width 32) with radial wave label smoothing at frequency 0.04. For visual clarity, we apply exponential averaging to all curves.

By choosing different functions $S$, we can probe nuances of spectral bias. Typically (inspired by Rahaman et al. [35]) we choose a radial wave: $S(\mathbf{X}) = \alpha(1 + \sin(2\pi f(\|\mathbf{X}\| - \mathbb{E}_{\mathcal{D}}\|\mathbf{X}\|)))$, where $\|\mathbf{X}\|$ denotes the Euclidean norm of the vectorized image, and $\alpha \in [0, 0.5]$ to ensure that $S(\mathbf{X}) \in [0, 1]$. We can vary the frequency $f$ to understand spectral bias at this global scale. We can also choose more targeted functions $S$; for instance, if $\mathbf{V}$ is a direction of interest through the space of images (*i.e.* $\mathbf{V}$ is an image-shaped vector of unit norm), we can construct $S(\mathbf{X}) = \alpha(1 + \sin(2\pi f \langle \mathbf{X}, \mathbf{V} \rangle))$. This allows us to vary both the frequency and direction of the noise function, to understand models' learnability along different directions through image space.

However, label smoothing has a few limitations. Because it involves perturbing the training labels, it cannot directly study the interaction between the training dataset and spectral bias, such as the effects of data augmentation. It also requires retraining each model from scratch, with substantial investment of time and computational resources. We address these limitations with our linear interpolation technique in the next section.

## 3.2 Linear Interpolation

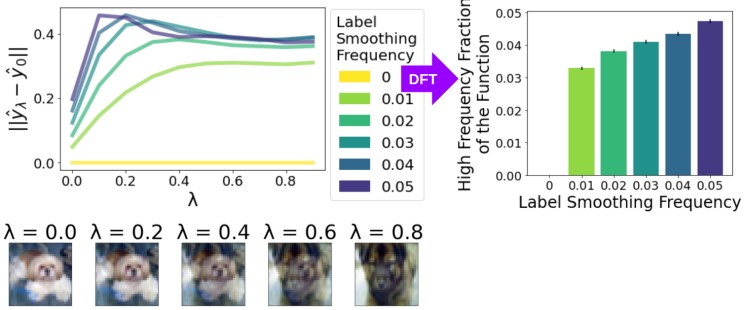

Figure 2: **Linear interpolation measurements on functions of varying frequency.** *Left*: Interpolation between images of the same class on an oracle function: the true one-hot label perturbed by radial wave label smoothing of variable frequency. As the oracle function increases in frequency, the interpolating paths become less smooth. *Right*: Summary of this interpolation experiment via a discrete Fourier transform; as the oracle frequency increases, so does the proportion of the DFT magnitude allocated to the high frequency components. In all interpolation figures, error bars report the standard error of the mean high frequency fraction over all the interpolating paths measured.

To complement the label smoothing approach, we also propose a methodology based on linear interpolation between validation images. Although we cannot take dense, regularly-spaced samples that cover the entire input (image) space, we can sample along specific paths and thereby glean glimpses into the spectral content of the learned function.

We consider two types of paths: those between images from the same class, and those between images from different classes. When considering paths between CIFAR-10 images in the same class, we choose 200 random, distinct pairs of validation images from each of the 10 CIFAR-10 classes and average over the paths defined by these pairs. We do the same for ImageNet, choosing 50 random, distinct paths for each of the 1000 ImageNet classes. When interpolating between CIFAR-10 classes, we choose 200 random, distinct validation image pairs from each of the 45 (10 choose 2) pairs of distinct classes, and average over the resulting paths; for ImageNet we choose 1000 random, distinct between-class paths, ensuring that each class is represented in exactly two paths.

For each pair of images $(\mathbf{X}_0, \mathbf{X}_1)$, we vary $\lambda \in [0, 1]$ to trace out a path, where each image in the path is given by $\mathbf{X}_\lambda = \lambda \mathbf{X}_1 + (1 - \lambda)\mathbf{X}_0$. We choose $\lambda$s so that the distance between adjacent images on the path is constant; each path may produce a different number of samples depending on the total distance between $\mathbf{X}_0$ and $\mathbf{X}_1$. Figure 2 *(Bottom Left)* shows an illustrative example of two CIFAR-10 images and a corresponding interpolation path between them. For our experiments, we typically have 50 to 100 images along the interpolation path.

**Relationship between interpolation and label smoothing experiments.** To validate our interpolation methodology, we consider the oracle function that maps directly from the interpolated image $\mathbf{X}_\lambda$ to the noised label used in our label smoothing experiments. We only consider within-class paths for this experiment, so that the original one-hot label (before smoothing) is fixed across the entire path, and the only change in the oracle function along the path is due to the noise function used in label smoothing. By increasing the frequency of the noise function, we create oracle functions with higher frequencies. We then verify that our interpolation measurement indeed measures higher frequencies on the higher frequency oracle functions.

In Figure 2 *(Left)*, we show an example where we vary the label smoothing frequency from $f = 0$ to $f = 0.05$. We plot the value of $\lambda$ along the interpolating path on the x axis, and the norm of the difference between the model output (softmax probabilities) on the interpolated image $\hat{\boldsymbol{y}}_\lambda$ and the model output on the original image $\hat{\boldsymbol{y}}_0$ on the y axis. This difference norm always starts at zero (although we plot it by averaging many paths over buckets in $\lambda$, so the curves do not actually touch the axis), and tends to increase with $\lambda$ as the model predictions change along the interpolating path. The norm of the prediction difference allows us to visualize how smooth the function is along the interpolating path; smoother prediction difference norm indicates a lower frequency function.

To quantify the frequency of the oracle function along the interpolating path, we compute the per-class discrete Fourier transform (DFT). Letting $\{\mathbf{X}_{\lambda_t}\}_{t=1}^{T}$ denote the set of interpolated images, the DFT coefficient for frequency $f$ is given by $\hat{\mathbf{Y}}_f[m] = \sum_{t=1}^{T} \hat{\boldsymbol{y}}_{\lambda_t}[m] \exp(-i2\pi f t)$, where $i$ is the imaginary number and $m \in \{1, \ldots, M\}$ denotes the class. To summarize how the magnitudes of the DFT coefficients $\hat{\mathbf{Y}}_f$ are distributed between low and high frequencies, we average the coefficient magnitudes for each of the classes and compute the fraction of the total average coefficient magnitude that is allocated to frequencies above a threshold (0.05). In Figure 2 *(Right)* we show this summary metric averaged over all within-class interpolating paths for the oracle functions, along with its standard error. In Figure 2 *(Left)*, the higher frequency oracle functions have less smooth prediction norm differences, and in Figure 2 *(Right)*, they have a higher fraction of DFT coefficient magnitudes in the high frequencies.

### 3.3 Data and Models

We use the CIFAR-10 [23] dataset of low-resolution ($32 \times 32$) natural images from ten animal and object classes and the ImageNet [8] dataset of higher-resolution ($224 \times 224$) natural images from 1000 classes. We consider a diversity of model architectures on CIFAR-10 (6 total) and ImageNet (10 total). See Section A.1 for more details on the data processing, models, and their accuracies.

## 4 Results on CIFAR-10

Throughout this section we show representative examples from our CIFAR-10 experiments; full results (on all six models we tested) are included in Section A.3.

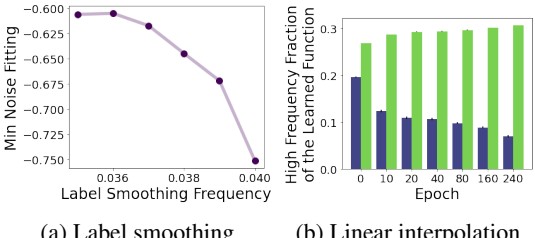

(a) Label smoothing  (b) Linear interpolation

Figure 3: **Modern CNNs have spatially-dependent spectral bias.** Higher min noise fitting denotes that the label smoothing noise function is learned more readily. *Left*: Min noise fitting for a `Shake-Shake96` model with variable-frequency radial wave label smoothing; lower frequencies are easier to learn. *Right*: Linear interpolation for a `ShakeShake96` model throughout training; as training proceeds, the learned function becomes lower-frequency within-class and higher-frequency between-class.

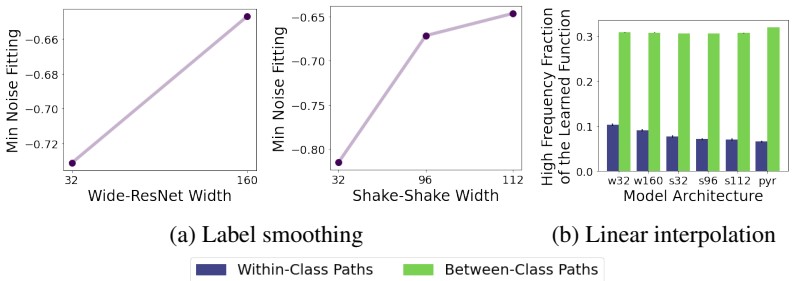

(a) Label smoothing  (b) Linear interpolation

■ Within-Class Paths  ■ Between-Class Paths

Figure 4: **Larger models learn high frequencies more readily; higher-accuracy models are higher-frequency between classes and lower-frequency within each class.** *Left*: Min noise fitting for Wide-ResNets and Shake-Shake models of variable width at frequency 0.039; the larger models learn this frequency more readily. *Right*: Linear interpolation frequencies for all six models we tested on CIFAR-10, ordered by test accuracy; higher-accuracy models are lower-frequency within-class and higher-frequency between-class.

## 4.1 Spectral Bias

We begin by applying our label smoothing methodology to find that modern image classification CNNs exhibit spectral bias, learning low frequency functions early in training and learning higher frequency functions as training proceeds. Note that Figure 3 *(Left)* shows spectral bias over a small but illustrative range of frequencies (the full range of frequencies we tested was 0 to 0.1); noise functions of sufficiently low frequency are fit almost immediately and noise functions of sufficiently high frequency are never learned during the 250 epochs of training we tested.

Using our linear interpolation methodology and distinguishing between within-class paths (in which both endpoint images are of the same class) and between-class paths (where the path must traverse a class boundary) in Figure 3 *(Right)*, we find that this spectral bias is highly variable over the domain of images. In particular, while the learned function does increase in frequency for between-class paths, it decreases in frequency for within-class paths, indicating that the model is learning to cluster and separate the different classes. Throughout our CIFAR-10 experiments, we find that higher-accuracy models have greater separation between the frequencies of their within-class and between-class paths, with lower-frequency behavior inside each class and higher-frequency behavior in between.

## 4.2 Spectral Bias and Model Architecture

Although all models we tested exhibit spectral bias, we found that the precise nature of the bias depends on the choice of model. For example, with all else fixed, increasing the width of a model decreases its spectral bias, enabling it to more readily fit noise functions of a given frequency. This trend is evident in Figure 4 for the Wide-ResNet family *(Left)* and the Shake-Shake family *(Center)*.

Although larger models learn high frequencies faster, we found that higher-accuracy models are not uniformly higher-frequency. Instead, using linear interpolation in Figure 4 *(Right)* we find that higher-

accuracy models (ordered left to right by increasing test accuracy) are higher-frequency along between-class paths but lower-frequency along within-class paths, perhaps evident of a better class clustering.

## 4.3 Sensitivity to Natural Image Directions

Label smoothing with a radial wave offers a convenient global picture of spectral bias, by replacing the radial wave with other noise functions we can use the same methodology to test more localized aspects of spectral bias. We take a step in this direction by considering the family of noise functions $S(\mathbf{X}; k) = \alpha(1 + \sin(2\pi f \langle \mathbf{X}, \mathbf{F}_k \rangle))$, where $\mathbf{F}_k$ is a diagonal Fourier basis image with frequency $k$ and the same dimensions as $\mathbf{X}$, visualized in Figure 5 *(Bottom)*.

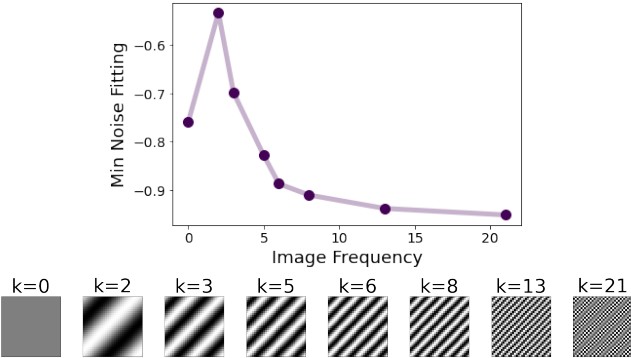

Figure 5: **Models are most sensitive to variations of low (but nonzero) image frequency.** *Top*: Effective noise fitting of a `WideResNet32` model with label smoothing of frequency 0.038 in various unit norm directions corresponding to Fourier basis images *(Bottom)* indexed by image frequency $k$ (scaled to [0, 1] for visualization).

We consider (a subset of) Fourier basis images because the Fourier spectra statistics of natural images are well studied (see, *e.g.* Tolhurst et al. [41]): natural images tend to be composed of Fourier basis images with amplitude proportional to their inverse spatial frequency. Indeed, in Figure 5 we find that `ShakeShake96` is more sensitive to label smoothing in low image frequency directions.

This finding is consistent with theoretical predictions of Basri et al. [2] that models learn faster in regions of higher density of training examples: since low image frequencies are more common in natural images, the effective sampling density enjoyed by a noise function is higher in these directions. It is also possible that this bias is a byproduct of the lower Fourier magnitudes of high frequency components in natural images [36], or is inherent to the convolutional model architecture regardless of the data distribution [33], or that a combination of multiple effects is at play. Determining the precise cause of this image frequency finding is an interesting direction for further study.

## 4.4 Spectral Bias and Training Data

In our CIFAR-10 interpolation experiments, we consistently observed that variations in training that improve accuracy create greater disparity between the frequencies of within-class and between-class paths, with lower frequencies within-class and higher frequencies between-class. This trend is evident in Figure 3 *(Right)* as a model trains for more epochs and in Figure 4 *(Right)* as the model architecture improves. In Figure 6 we find that the same trend also holds as we train on more data *(Left)*, apply more effective data augmentation *(Center)*, or apply self-distillation *(Right)*.

**Training Set Size.**    As shown in Figure 6 *(Left)* on `WideResNet160`, we find that increasing the number of training examples increases the difference in frequency content between within-class interpolating paths and between-class interpolating paths. Training with more data makes our CIFAR-10 models behave lower-frequency within each class but higher-frequency between classes, perhaps indicating that the models are learning a better clustering of the classes as they are provided with additional data.

**Data Augmentation.**    Data augmentation is a common strategy to increase the effective training set size without the expense of actually collecting additional examples. In Figure 6 *(Center)* we consider the effects of Mixup (with strength 0.1) [49], AutoAugment [6], and RandAugment [5], each of

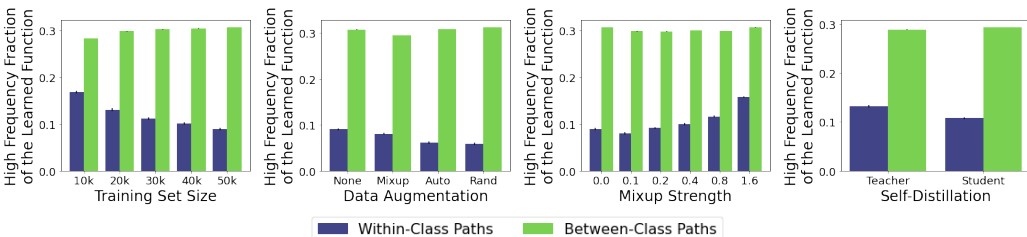

Figure 6: **Higher-accuracy models are lower-frequency within-class and higher-frequency between-class.** Test accuracy increases from left to right in each figure as we train `WideResNet160` with more data *(Left)*, better data augmentation *(Center Left)*, and self-distillation *(Right)*; in each case the frequency separation between within-class and between-class paths increases as accuracy improves. We also find that varying the Mixup strength *(Center Right)* within roughly the range 0.1 to 0.4 recommended by the original authors [49] tends to increase this frequency separation, but using stronger Mixup has the opposite effect.

which improves the final accuracy of the trained model (with RandAugment producing the most substantial benefits in our experiments, followed by AutoAugment and then Mixup). RandAugment and AutoAugment generate new images by applying geometric and lighting transformations, whereas Mixup linearly interpolates between pairs of existing images. Much like our experiments with training set size, we find that augmentations that improve accuracy more also produce greater frequency separation between within-class paths and between-class paths.

**Mixup Strength.** Mixup augmentation [49] perturbs each batch of training data by randomly pairing the examples and perturbing each example towards its partner by an interpolation amount $\lambda$ drawn from a symmetric beta distribution (with the same $\lambda$ used for all examples in the batch). We refer to the parameter of this beta distribution as the Mixup strength, as it controls the degree to which the augmented images tend to lie close to an original training image or close to the average of two training images. A parameter of 0 corresponds to no augmentation ($\lambda = 0$ or $\lambda = 1$, always using the original images), a parameter of 1 corresponds to the uniform distribution over $\lambda \in [0, 1]$, and a parameter of $\infty$ corresponds to $\lambda = 0.5$, the exact midpoint between a pair of training images. Figure 6 *(Center)* compares a range of Mixup strengths and finds that relatively low values that only slightly perturb the original images tend to produce greater frequency separation (and higher test accuracy), whereas perturbing the images too much with stronger Mixup has the opposite effect on both frequency separation and accuracy. Of note, the range of Mixup strengths that increase or do not affect frequency separation in our experiments are well aligned with those recommended by the original authors [49]: 0.1 to 0.4.

**Self-Distillation.** Finally, we consider the effect of self-distillation on the frequency content of the learned function. In self-distillation, a "teacher" model is trained with some form of strong regularization, in our case a combination of weight decay and early stopping based on training loss. A "student" model with the same architecture is then trained from scratch to fit the pseudolabels produced by the teacher Prior research [11] found that this procedure can train student models that outperform both their teachers and a baseline model trained as normal.

Prior research has also sought to understand the mechanism behind self-distillation. Mobahi et al. [28] finds that self-distillation acts as a regularizer by limiting the basis functions available to the student to learn. We complement their theoretical work with our interpolation experimental methodology in Figure 6 *(Right)*, where indeed we find that the student model learns a greater frequency separation between within-class and between-class paths than its teacher, while also achieving higher test accuracy. We conjecture that this effect is analogous to low-pass prefiltering common in digital signal processing: a high-frequency noise function that we cannot adequately sample is first smoothed (in this case by being approximated by a regularized teacher) and then it can be modeled via samples without further loss in fidelity. Without this prefiltering, our samples are inadequate to capture the complexity of the noise function, so we reconstruct an imperfect version corrupted by aliasing.

# 5  Results on ImageNet

Finally, we apply our linear interpolation methodology to study the frequency content of a range of pretrained models on ImageNet [8]. Here we show results on ResNet50; results on all ten models we tested on ImageNet are included in Section A.4.

## 5.1  Training Time

As we observed for CIFAR-10, our ImageNet models tend to increase the frequency gap between the two types of interpolating paths as training proceeds, with lower-frequencies within-class and higher-frequencies between-class. We visualize this trend for ResNet50 in Figure 7 *(Left)*. For our CIFAR-10 models, this increase in frequency difference between path types was achieved by a combination of increasing the average high frequency content on between-class paths and decreasing the average high frequency content on within-class paths. On ImageNet, we again find that average frequency content on between-class paths increases during training, but that within-class frequencies sometimes decrease during training (*e.g.* for ResNets) but sometimes increase (slower than between-class paths) during training (*e.g.* for CoATNets). We speculate that the trend of decreasing within-class frequency as training time progresses is less consistent on ImageNet than CIFAR-10 because, as we explore in the next section, ImageNet classes are more internally diverse than CIFAR-10 classes.

## 5.2  Class Coherence

In addition to containing $100\times$ as many classes as CIFAR-10, ImageNet classes can also be more internally diverse. This notion of within-class "consistency" is captured by C-Scores [18], which assign to each training example a value between 0 and 1 denoting how typical that example is. One way of capturing the internal consistency or diversity of a class is by studying the distribution of its C-Scores. In Figure 7 *(Right)* we plot the mean and standard deviation of these scores for each of the 1000 ImageNet classes in gray and highlight the 10 classes whose ResNet50 interpolating paths between validation images are most high-frequency (red stars) and least high-frequency (blue squares). We find that ResNet50 is lower-frequency on more internally consistent classes and higher-frequency on more diverse classes, supporting the intuition that more diverse classes require more complex functions.

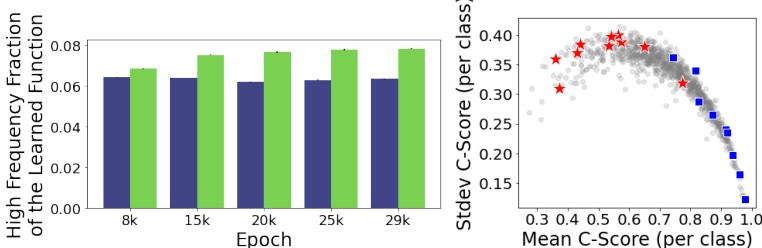

Figure 7: **Within vs. between-class frequency gap widens during training on ImageNet, and internally diverse classes have higher within-class function frequencies.** *Left*: The frequency separation between within-class and between-class paths for ResNet50 grows during ImageNet training. *Right*: Highlighted points show the 10 classes for which ResNet50 is most (red stars) and least (blue squares) high-frequency. Gray points show the distribution of C-Scores among all 1000 ImageNet classes.

# 6  Conclusions

In this paper, we introduced two methods to measure spectral bias in modern image classification neural networks, and applied these methods towards the central question:

*What kinds of function frequencies are needed for modern neural networks to generalize?*

Specifically, we applied these methods to examine the impact of a variety of training choices on the learned frequencies. On CIFAR-10, we found that higher-accuracy models typically have greater frequency separation between within-class and between-class paths, with lower frequencies within-class and higher frequencies between-class. This trend holds regardless of whether high accuracy

is achieved through longer training, choice of model architecture, increase in dataset size, data augmentation, or self-distillation. On ImageNet, we again found that this frequency separation increases during training, and noted that models are higher-frequency within more internally diverse classes. Our experimental methods offer a window onto the frequency structure of neural networks for image classification, and its relationship to model performance. In particular, although our primary goal is to improve our understanding of how neural networks generalize, our work is likely to have practical implications for improving generalization as well as robustness to image perturbations and distribution shifts. We hope that future work realizes these implications by applying our methods to study the spectral bias of robust models, and to induce model robustness directly, perhaps by incorporating our metrics as regularizers during training. Future work may also benefit from extending our methodology to interpolating paths that adhere more closely to the hypothesized manifold of natural images, such as those produced by generative networks [21, 40].

## Reproducibility Statement

Our CIFAR-10 code is available at https://github.com/google-research/google-research/tree/master/spectral_bias; for ImageNet we apply the same interpolation method to pre-cropped images and pretrained model checkpoints. Our work uses the publicly-available CIFAR-10 [23] and ImageNet [8] datasets.

## Acknowledgments

Many thanks to Yasaman Bahri, Ekin Dogus Cubuk, Hossein Mobahi, Michael Mozer, Maithra Raghu, Samuel Schoenholz, Jonathon Shlens, and Piotr Teterwak for productive conversations and helpful pointers.

SFK is also funded by NSF GRFP.

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
