# A  Appendix

## A.1  Data, Models, and Model Accuracies

Images are pixel-wise normalized by the mean and standard deviation of the training images for each dataset, and for ImageNet all images are center cropped and resized to $224 \times 224$; this preprocessing is done before any interpolating paths are constructed. For CIFAR-10, we consider six different convolutional neural networks: two Wide-ResNets [47] of different sizes (WideResNet32 and WideResNet160), three Shake-Shake regularized networks [12] of different sizes (ShakeShake32, ShakeShake96, and ShakeShake112), and PyramidNet, which uses the even stronger Shake-Drop regularization [45]. Our implementations are based on Cubuk et al. [6]; we use the same optimizer (stochastic gradient descent with momentum) and cosine learning rate schedule. We train without data augmentation (to ensure all models are trained on exactly the same examples), except for experiments that explicitly vary data augmentation. Without data augmentation, the test accuracies of our models are shown in Table 1.

| Model | CIFAR-10 Test Accuracy (%) |
|---|---|
| WideResNet32 | 89.4 |
| WideResNet160 | 90.2 |
| ShakeShake32 | 92.2 |
| ShakeShake96 | 93.5 |
| ShakeShake112 | 93.6 |
| PyramidNet | 95.8 |

Table 1: Test accuracies (in increasing order) of our six CIFAR-10 models, trained without data augmentation.

For ImageNet, we consider ten different pretrained models: three ResNets [16] of different sizes (ResNet50, ResNet101, and ResNet152), two Vision Transformers [10] of different sizes (ViT and ViTSmall), three distilled transformers [42] (DeiT, DeiTSmall, and DeiTClsToken), and two CoAtNets [7] (CoAtNet0 and CoAtNet0BF16). The top-1 classification accuracies of these models are presented in Table 2.

| Model | ImageNet Test Accuracy (%) |
|---|---|
| ViTSmall | 64.3 |
| ViT | 75.1 |
| ResNet50 | 76.1 |
| CoAtNet0BF16 | 78.4 |
| ResNet101 | 78.5 |
| ResNet152 | 79.0 |
| DeiTSmall | 80.1 |
| DeiTClsToken | 81.5 |
| DeiT | 81.7 |
| CoAtNet0 | 81.7 |

Table 2: Test accuracies of our ten pretrained ImageNet models, in increasing order.

## A.2  Linear Interpolation: Methodological Details

For each sampled path, we compute the discrete Fourier transform (DFT) of the prediction function along the path separately for each of the $M$ class predictions, take the (real) magnitude of the resulting (complex) DFT coefficients, and average them among the $M$ classes. We then compute the fraction of this averaged DFT magnitude that is allocated to the high frequency components, and aggregate over the many paths by calculating the mean high frequency fraction and its standard error. We use a simple threshold frequency of 0.05 for both CIFAR10 and ImageNet, although our paths are sampled with a distance of 1 for CIFAR10 and 7 for ImageNet (since ImageNet images are larger and thus farther apart). This precise threshold is not critical to our qualitative results, and different tradeoffs of frequency resolution and computational expense are possible.

**Limitations of linear interpolation.** Like all proxy measurements of spectral bias, linear interpolation is a coarse metric. We only measure the Fourier content of the learned function along specific paths, and aggregate results across many paths, across all classes, and only summarize frequencies into coarse "low" and "high" bins. An additional limitation is that the paths we define interpolate in pixel space, which means that there are images along our interpolating paths that are off the hypothesized manifold of natural images.

## A.3 CIFAR-10 full results

### A.3.1 Label Smoothing

Figure 8 shows the same results as Figure 3 *(Left)* on all six CIFAR-10 models we tested.

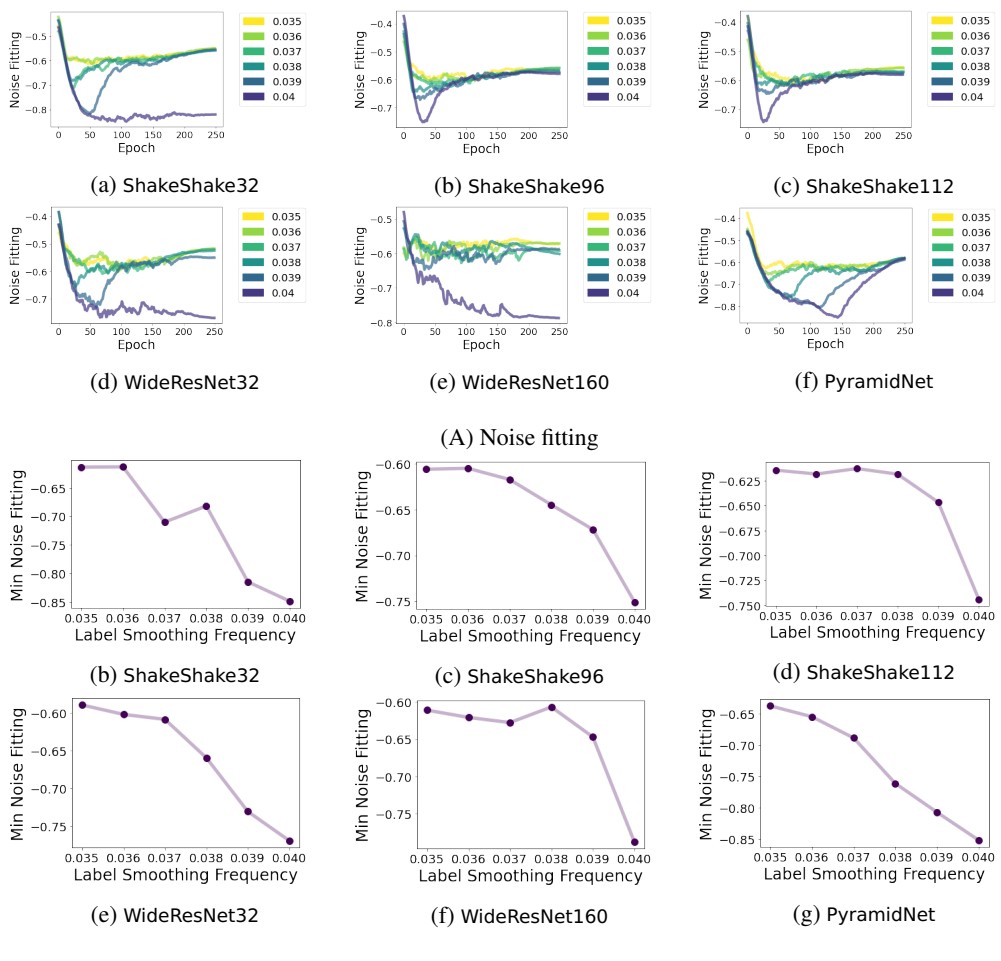

(A) Noise fitting

(B) Min noise fitting summaries

Figure 8: **All six CIFAR-10 models we tested exhibit spectral bias.** Here we show noise fitting when training each model with different frequencies of radial wave label smoothing.

### A.3.2 Sensitivity to Natural Image Directions

Figure 9 shows the same results as Figure 5 on all six CIFAR-10 models we tested.

### A.3.3 Agreement Between Label Smoothing and Linear Interpolation

**Varying the frequency of the radial wave label smoothing** When the frequency of the radial wave used for label smoothing increases, models take more time to fit the smoothing noise. Figure 3 *(Left)* and Figure 8 show this using label smoothing; the corresponding interpolation experiment is

shown in Figure 10. The target frequencies between 0.035 and 0.04 are close enough that, were the model to fit each perfectly, the interpolation curves would be nearly visually indistinguishable, as we can tell from Figure 2 *(Left)*. However, the model is actually smoother when trying to fit higher frequency label smoothing noise, because it fits this noise less well (in addition to fitting it later in training). We can see this effect, for instance, by noting that WideResNet32, WideResNet160, and ShakeShake32 fail to fit frequency 0.04, both in Figure 8 and Figure 10.

It is also worth noting that, in Figure 10 and repeatedly across our interpolation experiments, the learned function is smoother (lower-frequency) within-class and less smooth (higher-frequency) between-class. This is to be expected since the model must change predictions somewhere along the path between examples from different classes.

**Learning low frequencies first**   The label smoothing experiment presented in Figure 3 *(Left)* and Figure 8 shows that low frequency target functions are learned earlier than higher frequency targets; we can also confirm this finding using interpolation with model checkpoints saved at different epochs throughout training (without any label smoothing) in Figure 11. We see that spectral bias in image classification is more nuanced than simply increasing in frequency over time: as training proceeds, models become higher-frequency along between-class paths but lower-frequency along within-class

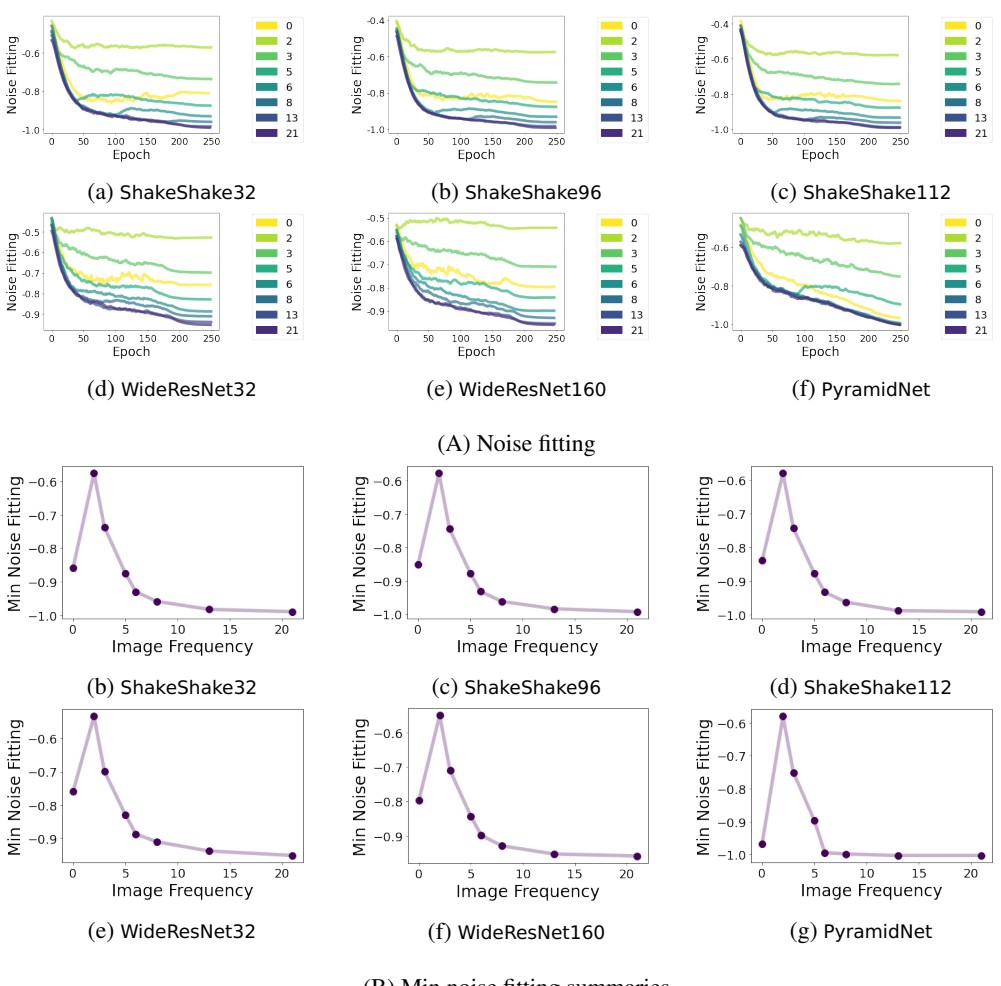

(A) Noise fitting

(B) Min noise fitting summaries

Figure 9: **All six CIFAR-10 models we tested exhibit sensitivity to variations of low (but nonzero) image frequency, which are dominant in natural images.** Here we show noise fitting when training each model with label smoothing of frequency 0.038 in various unit norm direction corresponding to Fourier basis images indexed by frequency $k$.

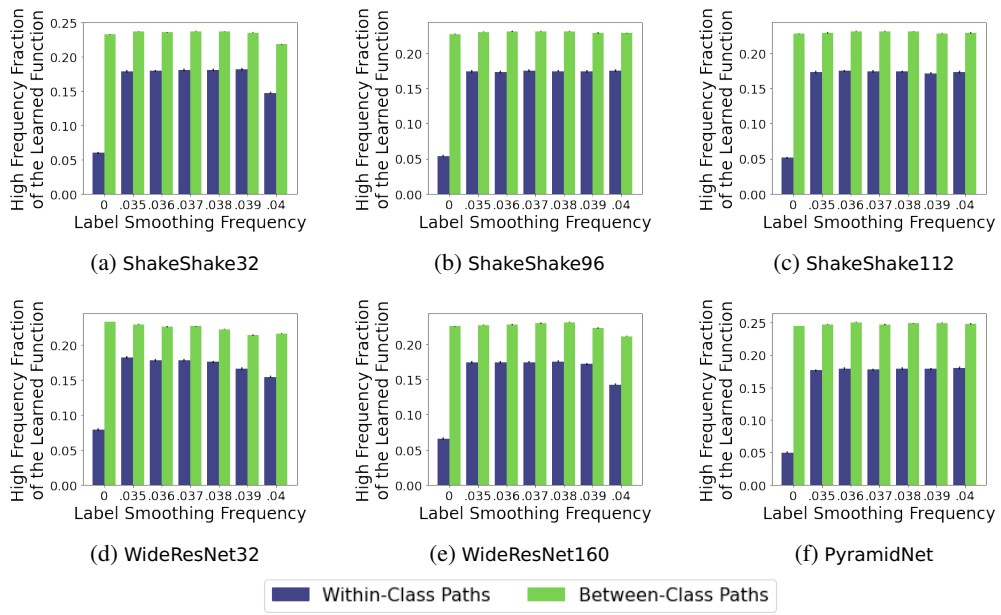

Figure 10: **Spectral bias is evident via interpolation when training with radial wave label smoothing at various frequencies.** Results here parallel those in Figure 8 but using the linear interpolation methodology.

paths. This may indicate that the model is learning an appropriate clustering of the data into its classes, and reducing the functional variation within each cluster.

### A.3.4 Weight Decay

Figure 12 uses the label smoothing methodology on our six CIFAR-10 models to show that weight decay delays the learning of high frequencies.

### A.3.5 Training Set Size

Figure 6 *(Left)* uses linear interpolation to show the within-class regularization (frequency reduction) effect of increasing dataset size for `WideResNet160`. Figure 13 shows the same effect on all six CIFAR-10 models we tested.

### A.3.6 Data Augmentation

Figure 6 *(Center)* uses linear interpolation to study the effect of common data augmentation procedures on the learned frequencies for `WideResNet160`. Figure 14 shows the same experiment on all six models we tested.

**Mixup Strength** Figure 6 *(Center)* uses linear interpolation to show that training with Mixup data augmentation [49] causes `WideResNet160` to learn a within-class lower-frequency function, but too much Mixup can produce a higher-frequency function within-class. Figure 15 repeats the experiment on all six CIFAR-10 models we tested.

### A.3.7 Self-Distillation

Figure 6 *(Right)* uses linear interpolation to show that self-distillation with `ShakeShake96` produces a student model that is lower-frequency within-class (and slightly higher-frequency between-class) than its teacher, and has higher validation accuracy than the teacher. Figure 16 shows the same result across all six CIFAR-10 models we tested.

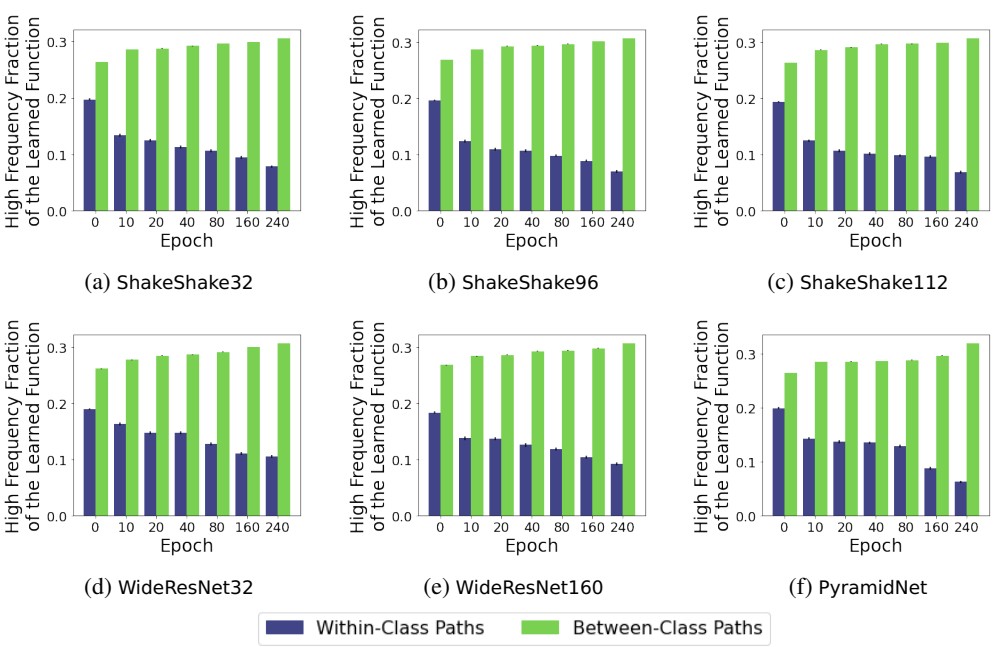

Figure 11: **All six CIFAR-10 models become higher-frequency between-class and lower-frequency within-class throughout training.** As accuracy improves, so does the frequency separation between within-class and between-class paths.

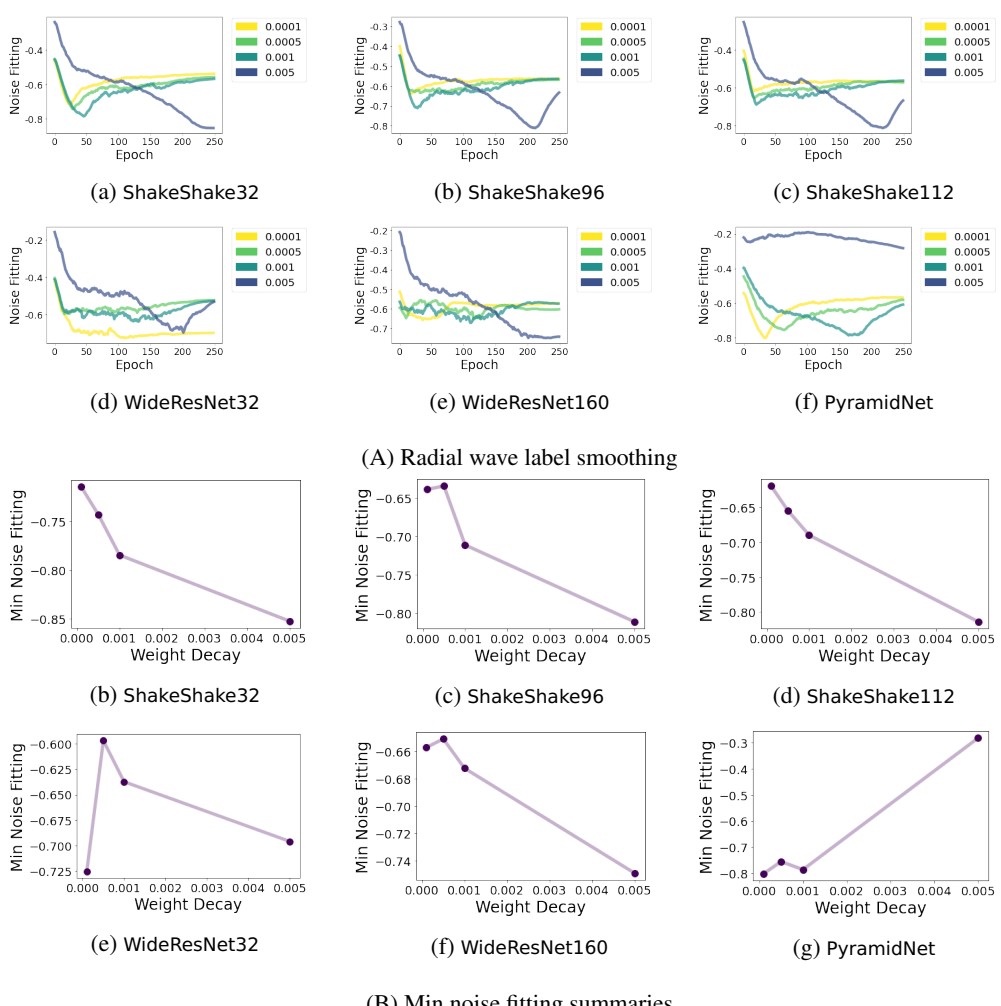

(A) Radial wave label smoothing

(B) Min noise fitting summaries

Figure 12: **All six CIFAR-10 models we tested are slower to learn high frequency target functions when trained with stronger weight decay.** Note that for PyramidNet, min noise fitting does not fully capture the results of noise fitting; the "dips" are clearly separated across epochs, with higher weight decay inducing delayed dips, even if they achieve similar minimum noise fitting values.

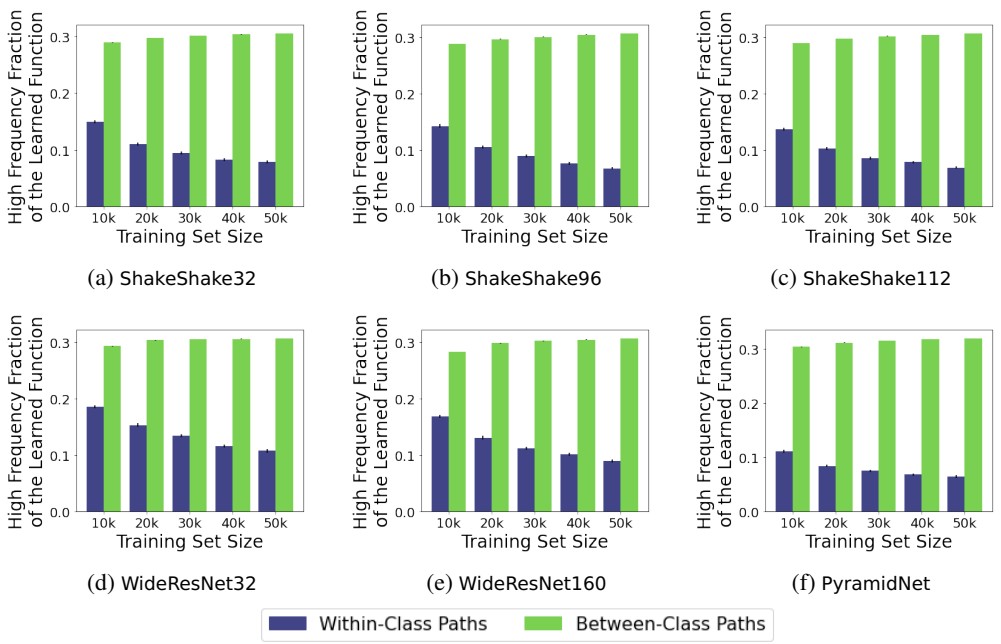

Figure 13: **For all six CIFAR-10 models we tested, increasing the number of training examples decreases the function frequency along within-class paths while slightly increasing the function frequency along between-class paths.**

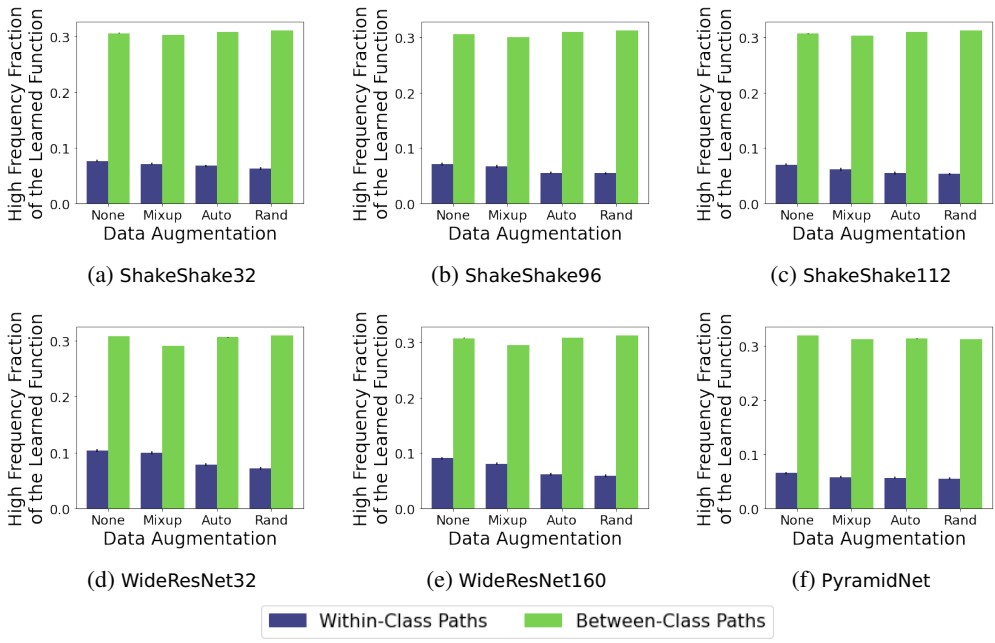

Figure 14: **On all six CIFAR-10 models, more effective data augmentation produces a model that is lower-frequency within-class.** In each figure, model test accuracy increases from left to right, as does the frequency separation between within-class and between-class paths. This trend parallels what we observe when increasing the number of training examples.

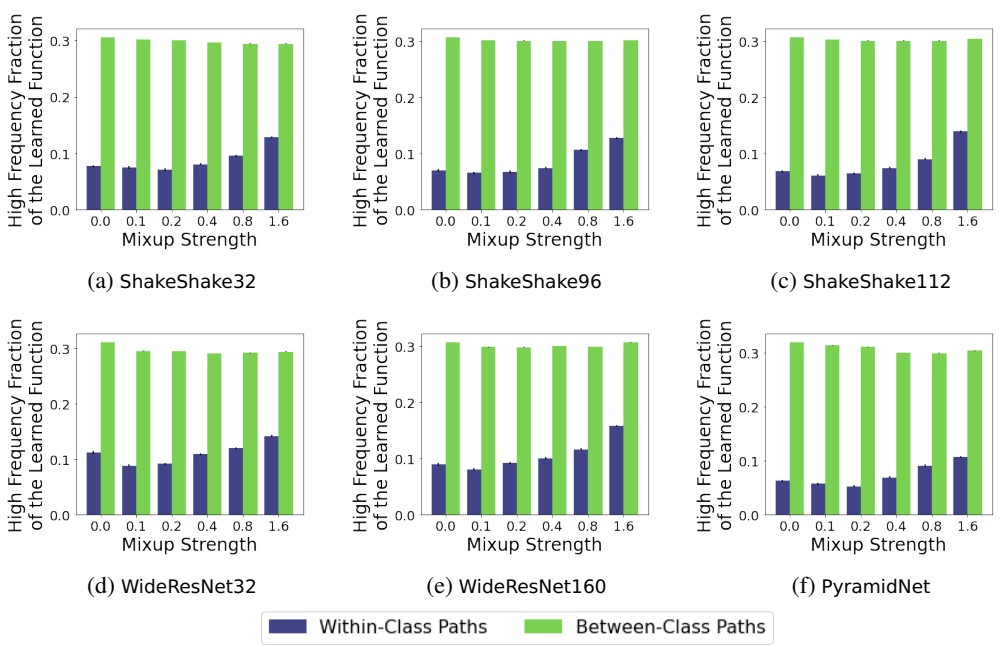

Figure 15: **On all six CIFAR10 models, modest Mixup augmentation produces a within-class lower-frequency learned function, but Mixup that is too strong can induce higher frequencies within-class.**

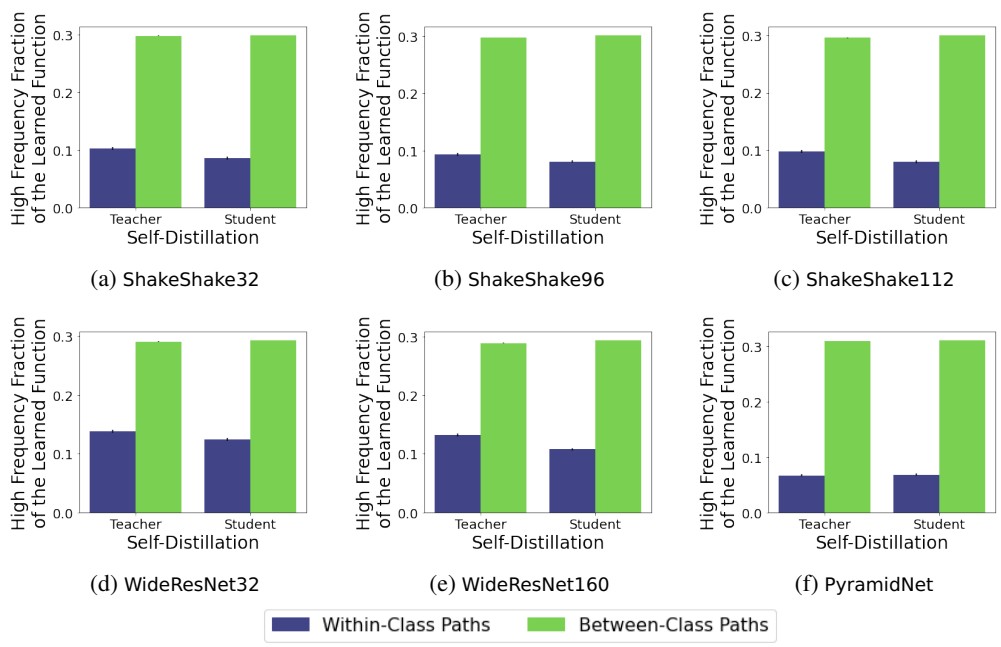

Figure 16: **In our CIFAR-10 interpolation experiments, self-distillation produces a student that is lower-frequency within-class, and has higher test accuracy, compared to its teacher.**

## A.4 ImageNet Full Results

### A.4.1 Training Time

Figure 7 *(Left)* shows that ResNet50 increases in between-class frequency as training proceeds, while within-class frequencies remain constant or slightly decrease, causing increasing frequency separation between these two types of paths. In Figure 17 we show the same experiment for all ten ImageNet models we tested. We find that the overall frequency trends vary between models, but all models increase in frequency separation between path types during training.

### A.4.2 Class Coherence

Figure 7 *(Right)* shows that ResNet50 has higher frequencies along paths within more internally diverse classes. In Figure 18 we show the same experiment for all ten ImageNet models we tested. Interestingly, the two CoATNets exhibit no clear correlation between class diversity and function frequency, and also dramatically increase in function frequency during training (Figure 17), unlike other models.

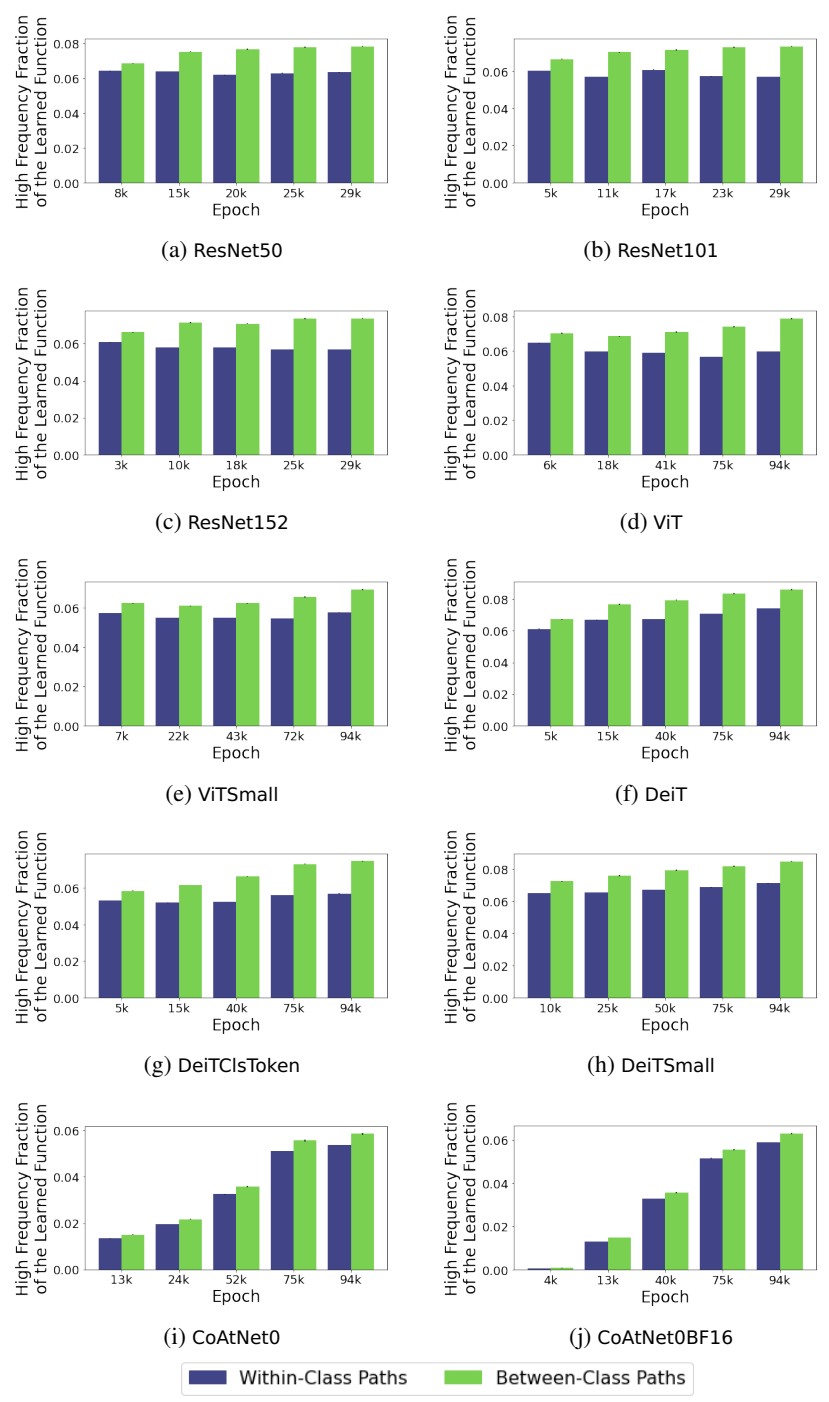

Figure 17: **On all ten ImageNet models, training longer increases the frequency separation between within-class and between-class paths.**

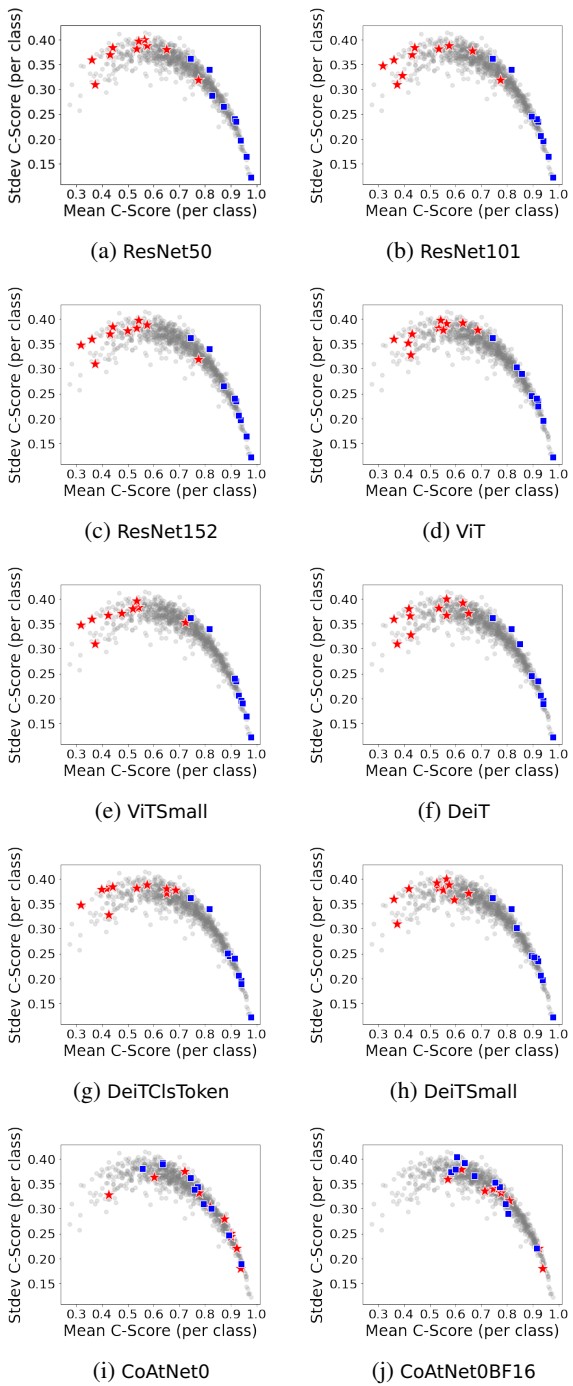

Figure 18: **On most of our ImageNet models, paths within more diverse classes have higher function frequencies.** Red stars show the 10 classes with the largest fraction of high-frequency content along within-class paths; blue squares show the 10 classes with the lowest fraction of high-frequency content along within-class paths. Note that CoAtNet models seem to deviate from this trend, with no clear relation between C-score and function frequency; understanding why is an interesting question for future work.