# OpenReview forum: "Spectral Bias in Practice: The Role of Function Frequency in Generalization"
_NeurIPS.cc/2022/Conference — NeurIPS 2022 Accept_

### Official Review · Reviewer_KubS · 2022-07-08

**Rating:** 6
**Confidence:** 4
**Soundness:** 3 good
**Presentation:** 2 fair
**Contribution:** 3 good

**Summary:**

This paper investigates the generalization of neural networks by investigating the spectra of the learned functions. While existing literature focuses on simple architectures like 2-layer MLPs, this work studies spectral bias in modern image classification networks. Two methodologies/metrics are developed for analyzing the spectral bias: i) label noise to enable measuring spectral bias in multi-class classification via label smoothing and ii) measuring the smoothness of a learned function via linear interpolation between test examples.The key finding is that high-accuracy models learn a function that is high frequency in regions between different image classes but low-frequency within each class. This finding is demonstrated to be valid under different settings, e.g., for different model architectures and training strategies.

**Questions:**

- It seems that noise fitting happens when the validation loss starts to plateau (Figure 1). A simple baseline would hence be to simply determine the noise fitting point via the slope of the validation loss. It would be interesting to see if this is an equally good measure to justify the complexity of perturbing labels and training models from scratch.
- Why does sampling along paths between images rather than in regions surrounding each image allow for a more global measurement of spectral bias (L.93)? Could the methodology by Zhang et al. also be used as a metric in this setting? If not, please explain why?
- L.171 why is the distance between adjacent images kept constant as opposed to the number of samples?
- How do the results from section 4.2 relate to the results for MLPs of Rahaman et al (On the spectral bias of neural networks)?
- The findings of Basri et al., that the learned function will be higher frequency in regions with denser samples (L.80), seem to contradict the finding of this work that the learned function of high-accuracy models is low-frequency within each class. Could you clarify how these findings can be aligned?
- Could you give an intuition for the formula for label smoothing in L.129?


**Limitations:**

The limitations were addressed adequately.

**Strengths And Weaknesses:**

Strengths:

- The paper is overall well-written and easy to follow.
- The proposed metrics/tools for analyzing spectral bias in more complex classification architectures are well-motivated and ablated. While the proposed metrics appear not completely novel (see weaknesses), applying these metrics to quantify spectral bias in neural networks is a valuable contribution and might be useful tools for future works in this research area.
- Generally, I found the paper insightful and interesting to read.


Weaknesses:
- The paper aims to explain the generalization ability of neural networks by investigating their spectral bias. However, there are no experiments in the paper that actually demonstrate that the proposed metrics are indeed linked to better generalization of the network (i.e. there is no metric to report generalization ability in the experiments).\
- The proposed label smoothing appears very similar to the label-smoothing regularization proposed in [37] and it is not clear enough how it differs.
- The proposed linear interpolation approach appears related to the Perceptual Path Length (PPL) regularization proposed in [1]. Please add this to the related work and discuss how PPL differs from the proposed linear interpolation.
- The terminology of ‘noise’ and ‘smoothing’ is confusing. E.g, L.136 the *noisy* validation loss corresponding to the *smoothed* labels., L.179 “label smoothing noise”. L.181, increasing the ‘label smoothing frequency’ corresponds to adding higher frequencies, i.e. noise.
- The key insight, that the learned function is high-frequency between classes and lower frequency within classes, is not very surprising considering that it is measured by linearly interpolating between samples of different, randomly chosen classes. For randomly chosen classes, a path very likely traverses multiple class boundaries which would explain higher frequencies. I suggest rather formulating this more conservatively, i.e. as a ‘proof of concept’ by confirming an intuitional result, as opposed to ‘a key insight’. Nonetheless, the proposed tools allow to quantify and support this intuition.

Missing related work:
- [1] for PPL, mentioned above.
- Model sensitivity to image frequency: [2] finds that discriminative models are not sensitive to the frequency range but rather to the magnitude of the frequencies.
Please also consider adding a brief discussion on how their finding aligns with your findings on image frequency. Sensitivity to magnitude might also provide another explanation for the higher sensitivity to natural image directions (4.3).

[1] T. Karrs, S. Laine, M. Aittala, J. Hellsten, J. Lethinen, T. Aila. Analyzing and Improving the Image Quality of StyleGAN. CVPR 2020

[2] K. Schwarz, Y. Liao, A. Geiger. On the Frequency Bias of Generative Models. Neurips 2021


Misc:
- In many references the journal/conference is missing.
- Consider renaming S from ‘target function’ to ‘noise function’ to i) be consistent with Rahaman et al and ii) make the name more intuitive.

---

> ### Author Response · Authors · 2022-08-02
> **Response (part 1)**
>
> Thank you for your positive and constructive feedback. We respond to individual questions/concerns inline below.
>
> >The paper aims to explain the generalization ability of neural networks by investigating their spectral bias. However, there are no experiments in the paper that actually demonstrate that the proposed metrics are indeed linked to better generalization of the network (i.e. there is no metric to report generalization ability in the experiments).
>
> We are considering the standard notion of (in-distribution) generalization in machine learning, which refers to accuracy on a held out test set not seen during training. For example, in Figure 6 all models in each subfigure are arranged in increasing order of test accuracy. The same is true for Figure 3b and Figure 4b.
>
> >The proposed label smoothing appears very similar to the label-smoothing regularization proposed in [37] and it is not clear enough how it differs.
>
> The formula for label smoothing in line 129 (line 122 in the revision) is indeed identical to that from [37], and is not a claimed contribution of our work. Rather, our contribution is in using label smoothing not as a regularizer to improve training (as in [37]) but as a measurement methodology to study spectral bias. We design our particular smoothing functions S to probe different frequency responses, not to improve accuracy or reduce overconfidence. We have edited the text of Section 3.1 to make it more clear that our use of label smoothing to measure spectral bias is novel, but label smoothing itself is not.
>
> >The proposed linear interpolation approach appears related to the Perceptual Path Length (PPL) regularization proposed in [1]. Please add this to the related work and discuss how PPL differs from the proposed linear interpolation.
> [1] T. Karrs, S. Laine, M. Aittala, J. Hellsten, J. Lethinen, T. Aila. Analyzing and Improving the Image Quality of StyleGAN. CVPR 2020
>
> Thanks for the suggestion; we are not sure how best to include this citation. It seems that Perceptual Path Length is a regularization technique using embedded distances, whereas our proposed linear interpolation is a measurement technique using raw Euclidean distances. We don’t see a direct connection to our linear interpolation approach, but perhaps the reviewer intended the reference as relevant to our discussion of future work, possibly using paths that are closer to the image manifold? Please let us know if this, or something else, is what the reviewer had in mind, thanks!
>
> >The terminology of ‘noise’ and ‘smoothing’ is confusing. E.g, L.136 the noisy validation loss corresponding to the smoothed labels., L.179 “label smoothing noise”. L.181, increasing the ‘label smoothing frequency’ corresponds to adding higher frequencies, i.e. noise.
>
> We thank the reviewer for this constructive feedback, and have edited the paper to use “noise” consistently. We agree this wording is clearer.
>
> >The key insight, that the learned function is high-frequency between classes and lower frequency within classes, is not very surprising considering that it is measured by linearly interpolating between samples of different, randomly chosen classes. For randomly chosen classes, a path very likely traverses multiple class boundaries which would explain higher frequencies. I suggest rather formulating this more conservatively, i.e. as a ‘proof of concept’ by confirming an intuitional result, as opposed to ‘a key insight’. Nonetheless, the proposed tools allow to quantify and support this intuition.
>
> Another reviewer had a similar question about this intuition; please refer to our combined response. In short, we agree that our results serve largely to support and validate this intuition, and have reworded the paper’s introduction accordingly.
>
> >Missing related work: Model sensitivity to image frequency: [2] finds that discriminative models are not sensitive to the frequency range but rather to the magnitude of the frequencies. Please also consider adding a brief discussion on how their finding aligns with your findings on image frequency. Sensitivity to magnitude might also provide another explanation for the higher sensitivity to natural image directions (4.3).
> [2] K. Schwarz, Y. Liao, A. Geiger. On the Frequency Bias of Generative Models. Neurips 2021
>
> We thank the reviewer for the reference, and have added a discussion to the revised paper. Indeed, this is an additional possible explanation (and not entirely unrelated to those we propose in the original manuscript).
>
> >In many references the journal/conference is missing.
>
> We thank the reviewer for their careful reading of our bibliography. In some cases this is because the cited work is a preprint and has yet to appear in published proceedings, but for those papers that have appeared in a conference or journal we have updated the citation accordingly.

---

> > ### Author Response · Authors · 2022-08-02
> > **Response (part 2)**
> >
> > >Consider renaming S from ‘target function’ to ‘noise function’ to i) be consistent with Rahaman et al and ii) make the name more intuitive.
> >
> > We agree this might be a clearer wording, and have implemented the suggestion in the revision.
> >
> > >It seems that noise fitting happens when the validation loss starts to plateau (Figure 1). A simple baseline would hence be to simply determine the noise fitting point via the slope of the validation loss. It would be interesting to see if this is an equally good measure to justify the complexity of perturbing labels and training models from scratch.
> >
> > This is an interesting suggestion. Although it would appear to work in Figure 1, in Figure 8 (a through f) in the appendix we do not always see a plateau in noise fitting before it increases. In particular, for lower frequency noise functions (like the yellow and green curves in Figure 8 a-f) the increase in noisy validation loss occurs before the clean validation loss plateaus, causing the minimum noise fitting to differ from the clean validation loss plateau.
> >
> > >Why does sampling along paths between images rather than in regions surrounding each image allow for a more global measurement of spectral bias (L.93)? Could the methodology by Zhang et al. also be used as a metric in this setting? If not, please explain why?
> >
> > Both methods are valid and interesting, but they measure different aspects of model behavior. The reason we consider our linear interpolation method to be a more global measurement is because it considers model behavior far away from input images, whereas the local sampling method of Zhang et al. only probes model behavior near input images. For example, we could not use the method of Zhang et al. to compare within-class vs between-class behavior, because all the samples used in that method are near input images and thus none can be between-class.
> >
> > >L.171 why is the distance between adjacent images kept constant as opposed to the number of samples?
> >
> > We keep the sampling distance constant between adjacent images so that the DFTs of different paths are directly comparable, having the same maximum frequency measured. We also experimented with a path-relative notion of frequency (i.e. in terms of the fraction of the interpolation distance rather than the absolute distance) and observed qualitatively similar results, but chose to present the more consistent and rigorous notion of frequency based on constant Euclidean distance sampling.
> >
> > >How do the results from section 4.2 relate to the results for MLPs of Rahaman et al (On the spectral bias of neural networks)?
> >
> > Our results in section 4.2 support and extend those of Rahaman et al. In particular, in Figure 16 of Rahaman et al, it is shown that increasing the width of an MLP increases its high frequency capacity, and that this high frequency content also increases as training proceeds. In our Figure 4a we show the same result for model width, and in Figure 3b we show that, while overall model frequency increases during training, this behavior is driven by between-class paths (which occupy a larger proportion of the total input space) while frequency actually decreases throughout training along within-class paths.
> >
> > >The findings of Basri et al., that the learned function will be higher frequency in regions with denser samples (L.80), seem to contradict the finding of this work that the learned function of high-accuracy models is low-frequency within each class. Could you clarify how these findings can be aligned?
> >
> > Indeed Basri et al. show that the learned function will be higher frequency where sampling is denser, but this is only true (and only claimed) up to the maximum frequency present in the target function. If a model is trying to fit a constant function, for example, its frequency will not (and should not) increase no matter how densely the target function is sampled. In our setting of within-class paths, the target function is indeed constant, as we want the model to predict the same class along the entire path.
> >
> > >Could you give an intuition for the formula for label smoothing in L.129?
> >
> > Note that this formula is not novel, but is rather borrowed and repurposed from [37]. That said, we agree that an intuitive description improves readability, and have added it to the paper. The intuition is that we want to replace a one-hot label vector with a discrete probability distribution that places less than full probability on the target class (hence scaling the original one-hot label by 1-S) and then distributes the remaining probability equally among the other classes (hence the addition of 1/M). Label smoothing is often used to express label uncertainty and reduce model overconfidence, but here we use it simply as a mechanism to extend the sinusoidal label noise of [34] to the multiclass setting.

---

> > > ### Comment · Reviewer_KubS · 2022-08-08
> > > **Response**
> > >
> > > Dear authors,
> > >
> > > thank you for addressing my concerns and considering my suggestions.
> > > Just a clarification regarding perceptual path length: Perceptual path length measures the perceptual distance between generated images. The images are generated by applying a generator to linearly interpolated latent codes, which is indeed different to linearly interpolating in image space. So as you suggested in your answer, it is a different way to construct a path (I understand that this is not feasible in your setting, as you do not have a generative model). What is, to some extent, similar between the metrics is that they both measure how much the output of a neural network changes for images generated along some path. But I agree that it is sufficient to discuss this in the future work section as you suggested.

---

> > > > ### Author Response · Authors · 2022-08-08
> > > > **Thanks for the clarification**
> > > >
> > > > Thank you for the clarification; we will add this citation and discussion in our future work section.

---

### Official Review · Reviewer_orks · 2022-07-11

**Rating:** 7
**Confidence:** 5
**Soundness:** 4 excellent
**Presentation:** 1 poor
**Contribution:** 3 good

**Summary:**

This paper aims to improve how the spectral bias of neural networks is measured. To this end, the authors introduce two techniques: first based on label smoothing, and the second based on probing the network outputs along a linear path between two data samples. The end result is a number of tricks that make spectral bias a more useful diagnostic tool for analyzing neural nets (pre-trained or otherwise), as empirically demonstrated in the experiments.

**Questions:**

- Regarding the experiment described in Figure 5: do the authors have an intuition as to what would happen if the analysis were to be performed on earlier layers of the network?
- Can the technique described in Section 3.2 be related to the Projection-slice Theorem?
- How exactly is “high-frequency fraction” defined? Is it the sum of the magnitudes of the DFT for f larger than some threshold? If so, what is this threshold?

**Limitations:**

The authors have been upfront about the limitations.

**Strengths And Weaknesses:**

### Positive
+ The use of label smoothing (instead of adding sinusoids to binary targets as done in Rahaman et al) is sound. The purpose of adding sinusoids to the target in Rahaman et al. was to expose the model to spatially correlated noise, which this method gracefully extends to the multiclass setting.
+ The idea of using the vector V to locally probe spectral bias is neat. I imagine that setting V to the principle component vectors of the data can also be insightful.
+ The idea of using projections of the function also makes sense, and I appreciate the sanity check in Figure 2 right.
+ The experiments are well crafted do a good job at teasing out interesting aspects of spectral bias.

-------
### Not Positive
+ The presentation can be improved. For instance, it would make for a more pleasant reading if the authors were to clearly (mathematically or pictorially) express what they mean by “noise fitting” and “min noise fitting” (even when the caption of Figure 1 explains what they are). The y-axis of Figure 2 left can also be confusing. In general, I feel the paper would read better if the experiments are bundled with their description (instead of being split in sections 3 and 4), but this is a matter of taste.

---

> ### Author Response · Authors · 2022-08-02
> **Response**
>
> Thank you for your positive and constructive feedback. We respond to individual questions/concerns inline below.
>
> >The idea of using the vector V to locally probe spectral bias is neat. I imagine that setting V to the principle component vectors of the data can also be insightful.
>
> This is a good idea for followup work, and has already been partially explored in Xu et al [41].
>
> >The presentation can be improved. For instance, it would make for a more pleasant reading if the authors were to clearly (mathematically or pictorially) express what they mean by “noise fitting” and “min noise fitting” (even when the caption of Figure 1 explains what they are). The y-axis of Figure 2 left can also be confusing. In general, I feel the paper would read better if the experiments are bundled with their description (instead of being split in sections 3 and 4), but this is a matter of taste.
>
> We appreciate this constructive feedback on our presentation. Indeed noise fitting is a tricky concept to illustrate, although we define it mathematically on line 142 (line 136 in the revision). We agree that it makes for more convenient reading to place methodology and results near each other, but opted instead to group similar results near each other. For example, in Figure 4 we compare spectral bias of different model architectures using both of our measurement methods, so we must either introduce both methods before this figure (as in the current paper) or separate the two halves of the figure and discuss spectral bias with respect to model architecture in two different sections. As the reviewer states, this is a matter of taste and both options have pros and cons.
>
> >Regarding the experiment described in Figure 5: do the authors have an intuition as to what would happen if the analysis were to be performed on earlier layers of the network?
>
> This is an interesting question, although not the focus of this work (we are interested in the output behavior). Our intuition is that for a convolutional network with small kernel size, earlier layers might have diminished response to low image frequencies, due to their small receptive fields.
>
> >Can the technique described in Section 3.2 be related to the Projection-slice Theorem?
>
> Not exactly. The projection-slice theorem relates the DFT of a projection to the slice of a DFT. Here we are not doing any projections, although we do average among many paths (but not necessarily paths that are aligned in any way, as they would be in a projection). There is a partial connection but it is not precise because of this averaging of random paths rather than projecting.
>
> >How exactly is “high-frequency fraction” defined? Is it the sum of the magnitudes of the DFT for f larger than some threshold? If so, what is this threshold?
>
> In short, yes and 0.05. We provide additional detail in the combined response, as another reviewer had a similar question.

---

### Official Review · Reviewer_ap1u · 2022-07-11

**Rating:** 7
**Confidence:** 4
**Soundness:** 3 good
**Presentation:** 3 good
**Contribution:** 3 good

**Summary:**

This work seeks to evaluate how training design decisions affects the spectral bias of neural networks. Specifically, they first extend the label noise procedure of Rahaman et. al to the multi-class setting and analyze when the deep learning model fits the noise. They also propose a second method which uses linear interpolation between two images, and use this to distinguish between the spectral content for "within classes" or "between classes". The authors consider results on CIFAR10 and on ImageNet, and consider factors such as accuracy, model size, training set size, and data augmentation.

**Questions:**

- Are there better ways of interpolating between two images (for example, optimal transport methods) rather than simply in pixel space. I could imagine that interpolating in a naive way could cause images to be relatively out of distribution.

- What do the spectral biases look like for non-convolutional networks (such as ViTs). Do they look similar to CNNs?

- In Section 5.2, you found a difference in spectral bias for diverse vs homogeneous classes? Do you see any differences between "hard" (i.e., low confidence) or easy examples.

**Limitations:**

The authors discuss limitations of the label smoothing method in Section 3.1. It would be useful to move up the discussion of the interpolation method from appendix A.2. The authors do not discuss any negative societal impacts of their work.

**Strengths And Weaknesses:**

## Strengths
*Quality*: I thought that the experiments were quite thorough, and was particularly impressed that they covered so many design decisions (such as data augmentation, training set size, etc).

*Clarity*: I thought the paper was relatively easy to follow.

*Originality/Significance*: This paper seems like an important step for making analysis of spectral biases practical. The paper is thus relevant to the community.

## Weaknesses

*Significance:* Please include some intuition on what these spectral biases translate to for real applications. What kind of spectral bias is desirable (and why)? Should we prioritize design decisions that are higher frequency between classes and lower frequency within classes (or does this just happen to be true for well-generalizing models)?  A final impact section, discussing what the concrete implications of this work are, would be helpful.

*Clarity*: The section on the relationship between the interpolation and label smoothing classes could be more clearly written.

---

> ### Author Response · Authors · 2022-08-02
> **Response**
>
> Thank you for your positive and constructive feedback. We respond to individual questions/concerns inline below.
>
> > Significance: Please include some intuition on what these spectral biases translate to for real applications. What kind of spectral bias is desirable (and why)? Should we prioritize design decisions that are higher frequency between classes and lower frequency within classes (or does this just happen to be true for well-generalizing models)? A final impact section, discussing what the concrete implications of this work are, would be helpful.
>
> We thank the reviewer for the constructive suggestion to add a discussion of concrete implications or suggestions for practitioners based on our findings, and we have added a brief discussion in the conclusions section of our revised paper. In particular, although the primary goal of our paper is to probe the theoretical phenomenon of spectral bias and thereby improve our understanding of how neural networks generalize, our work is likely to have practical implications for improving generalization as well as robustness to image perturbations and distribution shifts. We hope that future work realizes these implications by applying our methods to study the spectral bias of robust models, and to induce model robustness directly, perhaps by incorporating our metrics as regularizers during training.
>
> >Clarity: The section on the relationship between the interpolation and label smoothing classes could be more clearly written.
>
> We appreciate this feedback and have edited the text of that section in the revised paper for clarity.
>
> >Are there better ways of interpolating between two images (for example, optimal transport methods) rather than simply in pixel space. I could imagine that interpolating in a naive way could cause images to be relatively out of distribution.
>
> Another reviewer had a similar question; we refer the reviewer to our combined response.
>
> >What do the spectral biases look like for non-convolutional networks (such as ViTs). Do they look similar to CNNs?
>
> This is a very interesting question, and one we partially address in Figures 17 and 18 in the appendix. In these figures, we repeat the experiments in Figure 7 (which uses a ResNet50 on ImageNet) using 9 additional ImageNet models including ViT, DeiT, and CoAtNet. The trend of increasing within vs between class frequency separation during training holds across all of these models, but the trend of higher frequency paths on more diverse classes does not hold for CoAtNets (although it does for ViTs). This is an indication that, while the observations we made on CNNs in our paper do often extend to other architectures, this is not always the case. Understanding why different architectures exhibit these similar and different behaviors is beyond the scope of the present paper, but an important direction for future application of our proposed methods.
>
> >In Section 5.2, you found a difference in spectral bias for diverse vs homogeneous classes? Do you see any differences between "hard" (i.e., low confidence) or easy examples.
>
> These are actually very similar questions (difficulty vs diversity), and it is not obvious how to address them separately (although we are open to suggestions). In Section 5.2 we compare classes based on their mean and standard deviation C-Scores (introduced in [18]). A C-Score measures the probability that, if an example were held out from training, it would still be classified correctly. This can be viewed as measuring how easy the example is (higher C-Score denotes easier example, usually with higher confidence), or as measuring how consistent or similar the example is compared to the other training examples from the same class (if other similar examples are present in the training data, then this example is more likely to be classified correctly even if it is not trained on directly). We find higher path frequencies on classes that are both harder and more diverse.

---

> > ### Comment · Reviewer_ap1u · 2022-08-08
> > **Response**
> >
> > Thank you for your response. I maintain my positive score

---

### Official Review · Reviewer_7SRK · 2022-07-15

**Rating:** 4
**Confidence:** 4
**Soundness:** 2 fair
**Presentation:** 3 good
**Contribution:** 2 fair

**Summary:**

This paper presents a study of trained neural networks in the frequency domain. Two methodological contributions are proposed:
 1) an extension to multiclass classification of the technique in Rahaman et al for probing the ability of a model to learn signals of varying frequency
 2) a method for estimating the Fourier components of trained networks along linear interpolation paths between examples
The empirical contributions of this paper are:
 3) the spectral bias also applies to CIFAR10/Imagenet ResNets (whereas in Rahaman et al. only simpler models were studied)
 4) trained networks are on average lower frequency on within-class paths than on between-class paths
     1) higher disparity between within-class and between-class examples correlates with better generalization performance
     2) a correlation between more diverse classes (measured by examples with low mean C-scores) and high within-class frequency and vice versa.
 5) a relationship between image frequency and function frequency where are low image frequency components are learned faster than high image frequency components

**Questions:**

**Technical flow in 1/** (l129): "retain a valid probability distribution". Assume that S(X) = -1 (as can be the case when using the sine function mentioned l146), then for the true class, \bar{y} = 2-1/M is greater than 1, which does not define a valid probability distribution. How are you then able to train a model using these labels ? Are you using the cross-entropy loss or any other loss function?

**Technical flaw in 2/ thus 4/, 4.1/ and 4/2** (figure 3 right) If I understand the experiment correctly then within class I want to learn a function y(lambda=0) = y(lambda=1) = onehot(true class) and between class a function y(lambda=0) = onehot(class of example 0) \neq y(lambda=1) = onehot(class of example 1). Then I am not surprised that within class the Fourier coefficients are equally spread (in fact, they probably converge to 0 as you increase the number of examples), whereas between class you basically learn a function that grows continuously from 0 to 1. As you average over many examples, I would not be surprised if this resembled a sigmoid function. Can you maybe directly plot the (averaged) function ? If I am correct this then renders your findings somewhat trivial, but I might be proven wrong.

What exactly are you plotting in all barplots with low/high frequency components? What threshold are you using for splitting between low and high frequencies?

In figure 2 left: why is ||y_lambda - y_0|| > 0 when lambda = 0? when lambda = 1 (images come from the same class)?

**Other major comments**
 - l183: Can you make more precise what "model output" means? From the context I think it is the probability, but it could also be the logit output.
 - in figure 5, can you motivate the use of frequency=0.038? How does the experiment vary when using other values?

**Minor comments**
 - l170-175: I find the choice of a linear path and l2 (not precised in the text?) distance between images questionable. As a failure case, imagine 2 images sharing the same background B1 (e.g. grass) but with 2 different foreground objects C1 and C2, then the l2 distance between this images will likely be smaller than two images with the same foreground object class C1 but 2 different backgrounds B1 and B2.
 - figure 3 right: can you make precise whether this plot uses training set examples or test set examples?


**Limitations:**

I don't think that such a paper should discuss negative societal impact.

**Strengths And Weaknesses:**

**Originality**: This paper is in the line of other papers studying the so-called *spectral bias*: the tendency of neural networks to learn low-frequency functions first. All points listed in the summary are  original as far as I can tell.

**Quality**: The paper lacks some details in experiments and notation, and I think (but I might be proven wrong in the rebuttal) that there are some technical flaws in 1/, 2/ and 4/.

**Clarity**: Due to (potential) technical flaws, and missing details regarding experiments, I found it difficult to really judge on the significance of the experiments and the validity of the claimed results.

**Significance**: I feel that the methodological contributions are only minor improvements upon existing techniques (Rahaman et al for 1/ and Zhang et al. for 2/). Empirical observation 3/ is an extension of the findings of Rahaman et al. which can also be considered a minor improvement. I have the feeling that points 4/, 4.1/ and 4.2/ have a technical flaw/are trivial. I did not fully grasp empirical result 5/ so I am unable to judge of its significance.

---

> ### Author Response · Authors · 2022-08-02
> **Response (part 1)**
>
> We greatly appreciate the reviewer’s careful reading of our paper; we believe we have addressed all of your questions and concerns inline below as well as in the revised paper. In particular, we explain that the “technical flaws” mentioned in the review are actually a typo in an equation (which did not affect our experiments, and has been corrected in the revision) and perhaps a miscommunication of error bars (which are so small they are easily missed). We hope these responses will enable the reviewer to better judge the significance of our contributions, which were well summarized in the summary portion of the review. If additional questions remain, please do not hesitate to follow up and we are happy to clarify anything, both in our discussion and in the paper.
>
> > Significance: I feel that the methodological contributions are only minor improvements upon existing techniques (Rahaman et al for 1/ and Zhang et al. for 2/). Empirical observation 3/ is an extension of the findings of Rahaman et al. which can also be considered a minor improvement. I have the feeling that points 4/, 4.1/ and 4.2/ have a technical flaw/are trivial. I did not fully grasp empirical result 5/ so I am unable to judge of its significance.
>
> It is true that our methodologies bear similarity (and intentionally so) to these related prior works; we propose small changes to these methods that yield large returns in terms of allowing us to study modern models on more challenging datasets, and illuminate different questions about these more realistic models and datasets. In particular, with respect to Zhang et al. note that our interpolation procedure enables comparing different types of paths and uncovering that models exhibit different spectral bias within-class vs. between-class; this analysis would not be possible using prior methods.
>
> With respect to our empirical observations using label smoothing and those of Rahaman et al., it is important to note that our results cover realistic models on a much more realistic dataset (multiclass CIFAR-10 vs. binary MNIST and synthetic data) and probe more nuanced and varied aspects of spectral bias (through different directions in image space, for models of varying width, and throughout training time vs. only the latter for Rahaman et al.). Our experiments therefore offer considerable added value beyond those in prior work.
>
> Please also note that the “technical flaw” the reviewer mentioned is a typo/missing detail in the definition of the noise function S; it has now been corrected in the revised paper and did not affect any of our results (the code was already correct, and provided in the supplement). We thank the reviewer for spotting this typo and bringing it to our attention.
>
> > Technical flow in 1/ (l129): "retain a valid probability distribution". Assume that S(X) = -1 (as can be the case when using the sine function mentioned l146), then for the true class, \bar{y} = 2-1/M is greater than 1, which does not define a valid probability distribution. How are you then able to train a model using these labels ? Are you using the cross-entropy loss or any other loss function?
>
> Thanks for bringing this to our attention–there is a detail missing from line 146 and line 150 (lines 140 and 144 in the revision) where we define our sinusoidal choices for the smoothing function S. We actually shift (by adding one) and scale (by multiplying by 0.5 or less) the sinusoid to ensure its range is within [0, 1], as stated in line 127 (line 120 in the revision) and necessary to produce a valid probability distribution via label smoothing. We then train with cross-entropy loss, as stated in line 92 (line 86 in the revision). The paper has been updated with this important detail.

---

> > ### Author Response · Authors · 2022-08-02
> > **Response (part 2)**
> >
> > > Technical flaw in 2/ thus 4/, 4.1/ and 4/2 (figure 3 right) If I understand the experiment correctly then within class I want to learn a function y(lambda=0) = y(lambda=1) = onehot(true class) and between class a function y(lambda=0) = onehot(class of example 0) \neq y(lambda=1) = onehot(class of example 1). Then I am not surprised that within class the Fourier coefficients are equally spread (in fact, they probably converge to 0 as you increase the number of examples), whereas between class you basically learn a function that grows continuously from 0 to 1. As you average over many examples, I would not be surprised if this resembled a sigmoid function. Can you maybe directly plot the (averaged) function ? If I am correct this then renders your findings somewhat trivial, but I might be proven wrong.
> >
> > Another reviewer had a similar question about this intuition and its relation to our findings; please refer to our combined response. Please also note that all of our barplots, including Figure 3 (right), contain error bars denoting the standard error of the mean high frequency fraction plotted. These error bars are quite small (you might need to zoom in to see them) and suggest that averaging over more paths would not meaningfully change the result.
> >
> > > What exactly are you plotting in all barplots with low/high frequency components? What threshold are you using for splitting between low and high frequencies?
> >
> > Another reviewer asked a similar question; please refer to our combined response. Our threshold is 0.05; this was stated in the appendix in the original paper and is now also in the main text.
> >
> > > In figure 2 left: why is ||y_lambda - y_0|| > 0 when lambda = 0? when lambda = 1 (images come from the same class)?
> >
> > In Figure 2, we are plotting oracle functions of varying frequency (defined by label smoothing with sinusoids of varying frequency); we are not plotting the original one-hot labels or the predictions of a model. When the label smoothing is of zero frequency (the yellow curve), indeed the result is a flat line at zero. When label smoothing is of nonzero frequency, the oracle function contains a different label vector at the two endpoints because of this label smoothing (this explains why we see ||y_lambda - y_0|| > 0 when lambda = 1) . The reason ||y_lambda - y_0|| appears positive when lambda = 0 for these cases is that the plot is generated by binning over lambda ranges and averaging over many paths within each bin; although the function is zero at lambda = 0 we do not have a bin that includes only lambda identically zero, so the first bin includes some nonzero lambda values with their corresponding nonzero ||y_lambda - y_0|| values. We have clarified this in the revised paper.
> >
> > > l183: Can you make more precise what "model output" means? From the context I think it is the probability, but it could also be the logit output.
> >
> > You are correct that we are referring to the softmax (probability) outputs. We have clarified this in the revised paper.
> >
> > > in figure 5, can you motivate the use of frequency=0.038? How does the experiment vary when using other values?
> >
> > Figure 8 in the appendix illustrates how the six CIFAR-10 models we tested respond to label smoothing noise of varying frequency, within the range 0.035 to 0.04. We chose this range because it illustrates the most interesting behavior; for lower frequencies the noise is often learned immediately (too quickly to be readily observed) and for higher frequencies the noise is typically never learned within the training time. We then chose 0.038 for the image frequency experiment because it is near the middle of this “interesting range”, but the result is consistent as long as the frequency chosen is one for which the model exhibits spectral bias at an observable time scale.

---

> > > ### Author Response · Authors · 2022-08-02
> > > **Response (part 3)**
> > >
> > > >l170-175: I find the choice of a linear path and l2 (not precised in the text?) distance between images questionable. As a failure case, imagine 2 images sharing the same background B1 (e.g. grass) but with 2 different foreground objects C1 and C2, then the l2 distance between this images will likely be smaller than two images with the same foreground object class C1 but 2 different backgrounds B1 and B2.
> > >
> > > Yes, we use l2 distance. It is true that the Euclidean distance between two images is a poor proxy for their perceptual or semantic difference, but we are not using Euclidean distance as a similarity measure. Rather, we use it only to have some standard space in which to define frequency rigorously and to compare frequency fairly among different paths. We also experimented with a path-relative notion of frequency (i.e. in terms of the fraction of the interpolation distance rather than the absolute distance) and observed qualitatively similar results, but chose to present the more consistent and rigorous notion of frequency based on Euclidean distance. Regarding the choice of linear interpolation, another reviewer had a similar question; we refer the reviewer to our combined response.
> > >
> > > > figure 3 right: can you make precise whether this plot uses training set examples or test set examples?
> > >
> > > All of our figures consider test examples, as stated on line 159 in the methods section.

---

> > > > ### Comment · Reviewer_7SRK · 2022-08-08
> > > > **Ackowledgement**
> > > >
> > > > Thanks for your answer, that addresses my minor comments, and the definition of S. I accordingly raised my score.
> > > >
> > > > I however still thinks that your empirical result about within/between-class might be trivial. I think a good way of ruling this out would be to plot the function (evaluated for several examples) rather than its decomposition in the Fourier domain.

---

> > > > > ### Author Response · Authors · 2022-08-08
> > > > > **Figures you requested**
> > > > >
> > > > > Thanks for the feedback and for increasing your score! Apologies for not including these requested figures in the original response; at https://imgur.com/Sv1yAII we have reproduced a version of Figure 6 (our main results figure using linear interpolation) where instead of plotting high frequency fraction we plot the averaged paths themselves, as requested. As you can see, these figures show the same findings as Figure 6, where paths that show more variation (higher derivatives) have correspondingly higher high frequency fractions. In general of course the paths do usually increase from left to right (since they are constrained to all start at zero), but they do not always follow the same shape and often have dramatically different slopes. We actually originally visualized our paths using this method in an earlier draft of the paper, but ultimately decided that summarizing paths by their high frequency fraction offered a more quantitative metric (including error bars) that was a great deal easier to parse visually.

---

> ### Author Response · Authors · 2022-08-09
> **Requested figures**
>
> Hi Reviewer 7SRK,
>
> We posted the following reply this morning but realized it is somewhat difficult to find because each reply is nested inside the previous. So we are re-posting here to make the requested figure easier to find. Thanks for your engagement with and consideration of our work.
>
> Thanks for the feedback and for increasing your score! Apologies for not including these requested figures in the original response; at https://imgur.com/Sv1yAII we have reproduced a version of Figure 6 (our main results figure using linear interpolation) where instead of plotting high frequency fraction we plot the averaged paths themselves, as requested. As you can see, these figures show the same findings as Figure 6, where paths that show more variation (higher derivatives) have correspondingly higher high frequency fractions. In general of course the paths do usually increase from left to right (since they are constrained to all start at zero), but they do not always follow the same shape and often have dramatically different slopes (and right endpoints). We actually originally visualized our paths using this method in an earlier draft of the paper, but ultimately decided that summarizing paths by their high frequency fraction offered a more quantitative metric (including error bars) that was a great deal easier to parse visually.

---

### Author Response · Authors · 2022-08-02
**Common response to all reviewers**

We deeply thank the reviewers for their careful reading of our paper and thoughtful feedback, both positive and constructive. Reviewer 7SRK detailed our contributions of measurement methodologies and experimental investigation of spectral bias and described them as “original.” Reviewer ap1u described our experiments as “quite thorough” and our contributions as “an important step” and “relevant to the community.” Reviewer orks described our methodology as “sound” and our experiments as “well crafted.” Reviewer KubS described our work as “insightful and interesting to read.” We are very grateful for these kind words, as well as for the constructive suggestions made by the reviewers, which we address here and in the revised paper.

Some of the reviewer questions or concerns were shared by multiple reviewers, so we offer a combined answer to those here and reviewer-specific responses below.

*Significance with respect to intuition (7SRK, KubS):*

The reviewers’ intuition is that we expect a “good” model to have very low (ideally zero) frequencies along within-class paths and necessarily higher frequencies along between-class paths, and indeed this same intuition led us to consider these two types of paths. Note, however, that it is not at all obvious a priori that a high-accuracy model will satisfy our mutual intuition of relative frequency content along these two types of paths. A high-accuracy model must have some nonzero frequencies along between-class paths (to classify each endpoint image correctly), but there is no a priori reason why it should not also have high frequencies along within-class paths (in between the two endpoints, where the images are not realistic and the model may predict anything). Showing that, at least for the models we tested, this is not the case (i.e. model predictions are usually consistent along within-class paths), is an important finding that lends some evidence to support our original intuition (as pointed out also by Reviewer KubS). We have reworded the paper (in the introduction, shortly before the contributions) to clarify that our results are in support of this intuition, and are not wholly unexpected.

*Choice of linear interpolation (7SRK, ap1u):*

Two reviewers asked about our choice of linear interpolation, rather than alternative interpolation schemes (e.g. using optimal transport) that produce images closer to the hypothesized manifold of natural images. We did consider using “image-manifold” paths (defined by a GAN projection using Teterwak et al. 2021 [1], example here: https://imgur.com/a/R0mwihy) rather than linear paths, but unfortunately were unable to find a method that produced plausible natural images along an entire path. We chose to stick with linear interpolation to avoid any confounding effects of a partial but imperfect attempt to remain on-manifold or in-distribution. That said, as our modeling of the image manifold improves we hope future work will reconsider this direction as an extension of our work, when truly realistic-looking interpolations are more tractable.

[1] Piotr Teterwak, Chiyuan Zhang, Dilip Krishnan, and Michael C. Mozer. “Understanding invariance via feedforward inversion of discriminatively trained classifiers.” ICML 2021.

*Details of how “high frequency fraction” is computed (7SRK, orks):*

The process for generating our barplots of high frequency fraction is illustrated in Figure 2 with a toy example, and explained in the last paragraph of Section 3.2. We take the DFT of the class probabilities (softmax predictions) for each class along the path, average the magnitudes among the classes, and compute the fraction of the total averaged DFT magnitude that is allocated to frequencies that exceed a frequency threshold. Our frequency threshold is 0.05, as stated on line 515 in the appendix (we also added this value to Section 3.2 in the main text).

---

### Author Response · Authors · 2022-08-08
**Discussion period closing**

Thank you all for your time and thoughtful reviews! As the discussion period comes to a close, we ask that reviewers consider the final, following points of discussion.

**Reviewer 7SRK**
Thank you for your careful reading of our paper, and especially for helping us find a typo in the definition of the function S. We believe that we have addressed all of your concerns in our common and individual responses and in our revised manuscript, and that our paper is improved by these edits. Since we have not yet received feedback on our revised paper and responses to your original review, could you please consider confirming that we have addressed your concerns and raising your score accordingly? We are also happy to answer any remaining questions.

*Update*: We have provided the figure you requested (https://imgur.com/Sv1yAII), showing that there is indeed noticeable variation between different types of (averaged) paths even without summarizing them in the Fourier domain. Our Fourier summaries of these paths show that these path differences are statistically significant, and we believe they are also easier to read. We hope this addresses your concern. Thanks for engaging with our work!

**Reviewer ap1u**
Thank you for your positive assessment of our work! Based on your suggestions, we have revised our paper for clarity in our discussion of the relationship between our label smoothing and linear interpolation experiments, and we have added an additional discussion of the practical implications of our results.

**Reviewer orks**
Thank you for your positive assessment of our work, and for confirming that you read our response to your review. We appreciate it!

**Reviewer KubS**
Thank you for your positive assessment and for reading our response. We appreciate and have implemented your suggestions regarding additional related work discussion, using the term “noise” more consistently, and citing the journal version of papers in our bibliography, and we believe our revised paper is improved by these edits.

Thank you all again for your thoughtful comments that have improved our paper.

---

### Meta-Review · Area_Chair_tKeQ · 2022-08-26

**Recommendation:** Accept
**Confidence:** Certain

**Metareview:**

The authors have largely convinced the reviewers (and definitely myself) of the merits of the paper after extensive and detailed rebuttal and discussion. I am happy to recommend acceptance.

**Award:**

No

---

### Decision · Program_Chairs · 2022-09-14

Accept